# LEARNING-AUGMENTED FREQUENT DIRECTIONS

**Anders Aamand**
University of Copenhagen
andersaamanda@gmail.com

**Justin Y. Chen**
MIT
justc@mit.edu

**Siddharth Gollapudi**
Independent
sgollapu@berkeley.edu

**Sandeep Silwal**
UW-Madison
silwal@cs.wisc.edu

**Hao WU**
University of Waterloo
hao.wu1@uwaterloo.ca

## ABSTRACT

An influential paper of Hsu et al. (ICLR'19) introduced the study of learning-augmented streaming algorithms in the context of frequency estimation. A fundamental problem in the streaming literature, the goal of frequency estimation is to approximate the number of occurrences of items appearing in a long stream of data using only a small amount of memory. Hsu et al. develop a natural framework to combine the worst-case guarantees of popular solutions such as CountMin and CountSketch with learned predictions of high frequency elements. They demonstrate that learning the underlying structure of data can be used to yield better streaming algorithms, both in theory and practice.

We simplify and generalize past work on learning-augmented frequency estimation. Our first contribution is a learning-augmented variant of the Misra-Gries algorithm which improves upon the error of learned CountMin and learned CountSketch and achieves the state-of-the-art performance of randomized algorithms (Aamand et al., NeurIPS'23) with a simpler, deterministic algorithm. Our second contribution is to adapt learning-augmentation to a high-dimensional generalization of frequency estimation corresponding to finding important directions (top singular vectors) of a matrix given its rows one-by-one in a stream. We analyze a learning-augmented variant of the Frequent Directions algorithm, extending the theoretical and empirical understanding of learned predictions to matrix streaming.

## 1 INTRODUCTION

Learning-augmented algorithms combine the worst-case analysis of traditional algorithm design with machine learning to exploit structure in the specific inputs on which the algorithm is deployed. A burgeoning line of work in this context has studied algorithms furnished with predictions given by domain experts or learned from past data. This general methodology has been applied to create input-optimized data structures (Kraska et al., 2018; Mitzenmacher, 2018), graph algorithms (Dinitz et al., 2021; Chen et al., 2022c), online algorithms (Lykouris & Vassilvitskii, 2021; Gollapudi & Panigrahi, 2019), streaming algorithms (Hsu et al., 2019; Jiang et al., 2020; Chen et al., 2022a) among many other applications[1]. Within the context of streaming algorithms, where the input arrives in an online fashion and the algorithm has too little memory to store everything, predictors can highlight data which are worth remembering. This intuition was formalized in an influential work of Hsu et al. (2019) in the context of frequency estimation, a fundamental streaming problem where the goal is to provide an estimate of how many times any element appeared in the stream.

Given access to a heavy-hitter oracle identifying the highest frequency elements, Hsu et al. (2019) give a natural framework where the heavy-hitters are counted exactly while the rest of the frequencies are approximated using standard algorithms such as CountMin (Cormode & Muthukrishnan, 2005) or CountSketch (Charikar et al., 2002). They study the setting where the frequencies follow a power law distribution, commonly seen in practice and therefore well-studied for frequency estimation (Cormode & Muthukrishnan, 2005; Metwally et al., 2005; Minton & Price, 2012). Given access

---

[1]There are hundreds of papers written on this topics. See the survey of Mitzenmacher & Vassilvitskii (2022) or the website https://algorithms-with-predictions.github.io/.

to an oracle which can recover the heaviest elements, they give improved error bounds where error is taken in expectation over the empirical distribution of frequencies. A sequence of follow-up works investigate how to learn good predictors (Du et al., 2021; Chen et al., 2022b), apply the results to other streaming models (Shahout et al., 2024), and give improved algorithms (Aamand et al., 2023).

| Algorithms | Weighted Error | Predictions? | Analysis |
|---|---|---|---|
| Frequent Direction | $\Theta\left( \frac{\left(\ln \frac{m}{\ln \frac{2d}{m}}\right) \cdot \ln \frac{d}{m}}{(\ln d)^2} \cdot \frac{\|\mathbf{A}\|_F^2}{m} \right)$ | No | Theorem 3.3 |
| Learned Frequent Direction | $\Theta\left( \frac{1}{(\ln d)^2} \cdot \frac{\|\mathbf{A}\|_F^2}{m} \right)$ | Yes | Theorem 3.4 |

Table 1: Error bounds for Frequent Directions with $n$ input vectors from the domain $\mathbb{R}^d$ using $m \times d$ words of memory, assuming that the matrix consisting of input vectors has singular value $\sigma_i^2 \propto 1/i$. The weighted error is defined by Equation (2).

In this work, we define and analyze the corresponding problem in the setting where each data point is a vector rather than an integer, and the goal is to find frequent directions rather than elements (capturing low-rank structure in the space spanned by the vectors). This setting of "matrix streaming" is an important tool in big data applications including image analysis, text processing, and numerical linear algebra. Low-rank approximations via SVD/PCA are ubiquitous in these applications, and streaming algorithms for this problem allow for memory-efficient estimation of these approximations. In the matrix context, we define a corresponding notion of expected error and power-law distributed data. We develop a learning-augmented streaming algorithm for the problem based on the Frequent Directions (FD) algorithm (Ghashami et al., 2016) and give a detailed theoretical analysis on the space/error tradeoffs of our algorithm given predictions of the important directions of the input matrix. Our framework captures and significantly generalizes that of frequency estimation in one dimension. When the input vectors are basis vectors, our algorithm corresponds to a learning-augmented version of the popular Misra-Gries (Misra & Gries, 1982) heavy hitters algorithm. In this special case of our model, our algorithm achieves state-of-the-art bounds for learning-augmented frequency estimation, matching that of Aamand et al. (2023). In contrast to prior work, we achieve this performance without specializing our algorithm for power-law distributed data.

We experimentally verify the performance of our learning-augmented algorithms on real data. Following prior work, we consider datasets containing numerous problem instances in a temporal order (each instance being either a sequence of items for frequency estimation or a sequence of matrices for matrix streaming). Using predictions trained on past data, we demonstrate the power of incorporating learned structure in our algorithms, achieving state-of-the-art performance in both settings.

**Our Contributions**

- We generalize the learning-augmented frequency estimation model to the matrix streaming setting: each stream element is a row vector of a matrix $\mathbf{A}$. We define corresponding notions of expected error and power-law distributed data with respect to the singular vectors and values of $\mathbf{A}$.

- In this setting, we develop and analyze a learning-augmented version of the Frequent Directions algorithm of Ghashami et al. (2016). Given predictions of the important directions (corresponding to the top right singular vectors of $\mathbf{A}$), we demonstrate an asymptotically better space/error tradeoff than the base algorithm without learning. See Table 1.

- As a special case of our setting and a corollary of our analysis, we bound the performance of learning-augmented Misra-Gries for frequency estimation. In the learning-augmented setting, past work has analyzed randomized algorithms CountMin and CountSketch as well as specialized variants (Hsu et al., 2019; Aamand et al., 2023). To our knowledge, no analysis has been done prior to our work for the popular, deterministic Misra-Gries algorithm. Our analysis shows that learned Misra-Gries achieves state-of-the-art learning-augmented frequency estimation bounds, without randomness or specializing the algorithm for Zipfian data. See Table 2.

- We empirically validate our theoretical results via experiments on real data. For matrix streaming, our learning-augmented Frequent Directions algorithm outperforms the non-learned version by 1-2 orders of magnitude on all datasets. For frequency estimation, our learned Misra-Gries algorithm achieves superior or competitive performance against the baselines.

| Algorithm | Weighted Error | Rand? | Pred? | Reference |
|---|---|---|---|---|
| CountMin | $\Theta\left(\frac{n}{m}\right)$ | Yes | No | (Aamand et al., 2023) |
| Learned CountMin | $\Theta\left(\frac{\left(\ln\frac{d}{m}\right)^2}{m}\frac{n}{(\ln d)^2}\right)$ | Yes | Yes | (Hsu et al., 2019) |
| CountSketch | $O\left(\frac{1}{m}\frac{n}{\ln d}\right)$ | Yes | No | (Aamand et al., 2023) |
| Learned CountSketch | $O\left(\frac{\ln\frac{d}{m}}{m}\frac{n}{(\ln d)^2}\right)$ | Yes | Yes | (Aamand et al., 2023) |
| CountSketch++ | $O\left(\frac{\ln m + \text{poly}(\ln\ln d)}{m}\frac{n}{(\ln d)^2}\right)$ | Yes | No | (Aamand et al., 2023) |
| Learned CountSketch ++ | $O\left(\frac{1}{m}\frac{n}{(\ln d)^2}\right)$ | Yes | Yes | (Aamand et al., 2023) |
| Misra-Gries | $\Theta\left(\left(\ln\frac{m}{\ln\frac{2d}{m}}\right)\frac{\ln\frac{d}{m}}{m}\frac{n}{(\ln d)^2}\right)$ | No | No | Theorem 3.1 |
| Learned Misra-Gries | $\Theta\left(\frac{1}{m}\frac{n}{(\ln d)^2}\right)$ | No | Yes | Theorem 3.2 |

Table 2: Error bounds for frequency estimation with $n$ input elements from the domain $[d]$ using $m$ words of memory, assuming that the frequency of element $i \in [d]$ follows $f(i) \propto 1/i$. The weighted error indicates that element $i$ is queried with a probability proportional to $1/i$.

**Related Work**    The learning-augmented frequency estimation problem was introduced in Hsu et al. (2019). They suggest the model of predicted frequencies and give the first analysis of learning-augmented CountMin and CountSketch with weighted error and Zipfian frequencies. Du et al. (2021) evaluate several choices for the loss functions to use to learn the frequency predictor and Chen et al. (2022b) develop a procedure to learn a good predictor itself with a streaming algorithm. Shahout et al. (2024) extend the model to sliding window streams where frequency estimation is restricted to recently appearing items. Shahout & Mitzenmacher (2024) analyze a learning-augmented version of the SpaceSaving algorithm (Metwally et al., 2005) which is a deterministic algorithm for frequency estimation, but, unlike our work, they do not give space/error tradeoffs comparable to Hsu et al. (2019). Aamand et al. (2023) give tight analysis for CountMin and CountSketch both with and without learned predictions in the setting of weighted error with Zipfian data. Furthermore, they develop a new algorithm based on the CountSketch, which we refer to as learning-augmented CountSketch++, which has better asymptotic and empirical performance.

Matrix sketching and low-rank approximations are ubiquitous in machine learning. The line of work most pertinent to our work is that on matrix streaming where rows arrive one-by-one, and in small space, the goal is to maintain a low-rank approximation of the full matrix. The Frequent Directions algorithm for the matrix streaming problem was introduced by Liberty (2013). Subsequent work of Ghashami & Phillips (2014a) and Woodruff (2014) refined the analysis and gave a matching lower bound. These works were joined and developed in Ghashami et al. (2016) with an even simpler analysis given by Liberty (2022).

A related line of work is on learning sketching matrices for low-rank approximation, studied in Indyk et al. (2019; 2021). Their goal is to learn a sketching matrix $\mathbf{S}$ with few rows so that the low-rank approximation of $\mathbf{A}$ can be recovered from $\mathbf{SA}$. The main guarantee is that the classical low-rank approximation algorithm of Clarkson & Woodruff (2013), which uses a random $\mathbf{S}$, can be augmented so that only half of its rows are random, while retaining worst-case error. The learned half of $\mathbf{S}$ can be optimized empirically, leading to a small sketch $\mathbf{SA}$ in practice. The difference between these works and us is that their overall procedure cannot be implemented in a single pass over the stream. We discuss other related works in Appendix A.

**Organization**    Section 2 delves into the necessary preliminaries for our algorithm. We define the problems of frequency estimation, and its natural higher dimensional version, introduce our notion of estimation error for these problems, and discuss the two related algorithms Misra-Gries and Frequent Directions for these problems. In Section 3, we introduce our learning-augmented versions of Misra-Gries and Frequent Directions. We also analyse the performance of learned Misra-Gries algorithms, postponing the the analysis of learned Frequent Directions to Appendix D. Section 4 presents our experiment results with extensive figures given in Appendix F.

## 2 PRELIMINARIES

**Frequency Estimation.** Let $n, d \in \mathbb{N}^+$, and consider a sequence $a_1, a_2, \ldots, a_n \in [d]$ arriving one by one. We are interested in the number of times each element in $[d]$ appears in the stream. Specifically, the frequency $f(i)$ of element $i \in [d]$ is defined as $f(i) \doteq |\{t \in [n] : a_t = i\}|$. Thus, $\sum_{i \in [d]} f(i) = n$. Given estimates $\tilde{f}(i)$ for each $, i \in [d]$, we focus on the following weighted estimation error (Hsu et al., 2019; Aamand et al., 2023): $\mathcal{E}rr \doteq \sum_{i \in [d]} \frac{f(i)}{n} \cdot \left| f(i) - \tilde{f}(i) \right|. \, (1)$

The weighted error assigns a weight to each element's estimation error proportional to its frequency, reflecting the intuition that frequent elements are queried more often than less frequent ones.

**Direction Frequency.** The frequency estimation problem has a natural high-dimensional extension. The input now consists of a stream of vectors $\mathbf{A}_1, \mathbf{A}_2, \ldots, \mathbf{A}_n \in \mathbb{R}^d$. For each unit vector $\vec{v} \in \mathbb{R}^d$, we define its "frequency," $f(\vec{v})$, as the sum of the squares of the projected lengths of each input vector onto $\vec{v}$. Specifically, let $\mathbf{A} \in \mathbb{R}^{n \times d}$ denote the matrix whose rows are $\mathbf{A}_1^T, \ldots, \mathbf{A}_n^T$. Then $f(\vec{v}) \doteq \|\mathbf{A}\vec{v}\|_2^2$.

To see the definition is a natural extension of the element frequency in the *frequency estimation* problem, suppose each input vector $\mathbf{A}_t$ is one of the standard basis vectors $\vec{e}_1, \ldots, \vec{e}_d$ in $\mathbb{R}^d$. Further, we restrict the frequency query vector $\vec{v}$ to be one of these standard basis vectors, i.e., $\vec{v} = \vec{e}_i$ for some $i \in [d]$. Then $f(\vec{e}_i) = \|\mathbf{A}\vec{e}_i\|_2^2 = \sum_{t \in [n]} \langle \mathbf{A}_t, \vec{e}_i \rangle^2 = \sum_{t \in [n]} \mathbb{1}_{[\mathbf{A}_t = \vec{e}_i]}$, which is simply the number of times $\vec{e}_i$ appears in $\mathbf{A}$.

*Estimation Error.* Consider an algorithm that can provide an estimate $\tilde{f}(\vec{v})$ of $f(\vec{v})$ for any unit vector $\vec{v} \in \mathbb{R}^d$. The estimation error of a single vector is given by $\left| f(\vec{v}) - \tilde{f}(\vec{v}) \right| = \left| \|\mathbf{A}\vec{v}\|_2^2 - \tilde{f}(\vec{v}) \right|$. Since the set of all unit vectors in $\mathbb{R}^d$ is uncountably infinite, we propose to study the following weighted error:

$$\mathcal{E}rr = \sum_{i \in [d]} \frac{\sigma_i^2}{\|A\|_F^2} \cdot \left| \|\mathbf{A}\vec{v}_i\|_2^2 - \tilde{f}(\vec{v}_i) \right|, \tag{2}$$

where $\sigma_1, \ldots, \sigma_d$ denote the singular values of $\mathbf{A}$, $\vec{v}_1, \ldots, \vec{v}_d$ are the corresponding right singular vectors, and $\|\mathbf{A}\|_F$ is its Frobenius norm.

To see how this generalizes Equation (1), assume again that the rows of $\mathbf{A}$ consist of standard basis vectors and that $f(\vec{e}_1) \geq f(\vec{e}_2) \geq \cdots \geq f(\vec{e}_d)$. In this case, it is straightforward to verify that $\sigma_i^2 = f(\vec{e}_i)$ and $\vec{v}_i = \vec{e}_i$ for all $i \in [d]$. Consequently, $\|\mathbf{A}\vec{v}_i\|_2^2 = f(\vec{e}_i)$, and $\|\mathbf{A}\|_F^2 = \sum_{i \in [d]} \sigma_i^2 = n$. Therefore, Equation (2) reduces to Equation (1) in this case. Moreover, for a specific class of algorithms, we can offer an alternative and intuitive interpretation of the weighted error.

**Lemma 2.1.** *For algorithms that estimate $\tilde{f}(\vec{v})$ by first constructing a matrix $\mathbf{B}$ and then applying the formula $\tilde{f}(\vec{v}) = \|\mathbf{B}\vec{v}\|_2^2$ such that $0 \leq \tilde{f}(\vec{v}) \leq f(\vec{v})$, the weighted error defined in Equation (2) satisfies $\mathcal{E}rr \propto \mathbb{E}_{\vec{v} \sim N(0, \mathbf{A}^T \mathbf{A})} \left[ \|\mathbf{A}\vec{v}\|_2^2 - \tilde{f}(\vec{v}) \right]. \, (3)$*

The conditions stated in the lemma apply to the Frequent Directions algorithm (Ghashami et al., 2016), discussed later in the section. The lemma asserts that the weighted error is proportional to the expected difference between $\|\mathbf{A}\vec{v}\|_2^2$ and $\tilde{f}(\vec{v})$, where $\vec{v}$ is sampled from a multivariate normal distribution with mean 0 and covariance matrix $\mathbf{A}^T \mathbf{A}$. The proof is included in the Appendix B.

**Zipfian Distribution.** We follows the assumption that in the frequency estimation problem, the element frequencies follow a Zipfian distribution (Hsu et al., 2019; Aamand et al., 2023), i.e., $f(i) \propto 1/i \quad \forall i \in [d]$. Naturally, for the high dimensional counterpart, we assume that $\sigma_i^2 \propto 1/i$.

**Misra-Gries and Frequent Directions Algorithms.** The Misra-Gries algorithm (Misra & Gries, 1982) is a well-known algorithm developed for frequency estimation in the streaming setting with limited memory. Its high-dimensional counterpart is the Frequent Directions algorithm (Ghashami et al., 2016). We focus on presenting the Frequent Directions algorithm here along with a brief explanation of how Misra-Gries can be derived from it.

The algorithm is described in Algorithm 1. The matrix $\mathbf{B}$ created during the initialization phase can be viewed as an array of $m$ buckets, where each bucket can store a vector in $\mathbb{R}^d$. As each input vector $\mathbf{A}_i$ arrives, the algorithm updates $\mathbf{B}$ using an "update" procedure, inserting $\mathbf{A}_i^T$ into the first available bucket in $\mathbf{B}$. If $\mathbf{B}$ is full, additional operations are triggered (Lines 7 - 10): essentially, the algorithm performs a singular value decomposition (SVD) of $\mathbf{B}$, such that $\mathbf{B} = \sum_{j \in [d]} \sigma_j^{(i)} \cdot \vec{u}_j^{(i)} \left( \vec{v}_j^{(i)} \right)^T$, where $\vec{u}_j^{(i)}$ and $\vec{v}_j^{(i)}$ are the columns of matrices $\mathbf{U}^{(i)}$ and $\mathbf{V}^{(i)}$, respectively, and $\sigma_j^{(i)}$ are the diagonal entries of $\mathbf{\Sigma}^{(i)}$. The algorithm then retains only the first $\tau - 1$ right singular vectors, $\vec{v}_1^{(i)}, \ldots, \vec{v}_{\tau-1}^{(i)}$, scaled by the factors $\left( (\sigma_1^{(i)})^2 - (\sigma_\tau^{(i)})^2 \right)^{1/2}, \ldots, \left( (\sigma_{\tau-1}^{(i)})^2 - (\sigma_\tau^{(i)})^2 \right)^{1/2}$ respectively.

---

**Algorithm 1** Frequent Direction $\mathcal{A}_{FD}$

---

1: **Procedure** INITIALIZATION
2:     **Input:** sketch parameters $m, \tau, d \in \mathbb{N}^+$, s.t., $\tau \le m \le d$
3:     Reserve $m \times d$ space for an empty matrix $\mathbf{B}$

4: **Procedure** UPDATE
5:     **Input:** an input vector $\mathbf{A}_i \in \mathbb{R}^d$
6:     $\mathbf{B} \leftarrow [\mathbf{B}; \mathbf{A}_i^T]$ matrix obtained by appending $\mathbf{A}_i^T$ after the last row $\mathbf{B}$
7:     **if** $\mathbf{B}$ has $m$ rows **then**
8:         $\mathbf{U}^{(i)}, \mathbf{\Sigma}^{(i)}, \mathbf{V}^{(i)} \leftarrow \text{SVD}(\mathbf{B})$
9:         $\overline{\mathbf{\Sigma}^{(i)}} \leftarrow \sqrt{\max\{\mathbf{\Sigma}^{(i)^2} - (\sigma_\tau^{(i)})^2 \boldsymbol{I}, 0\}}$, where $\sigma_\tau^{(i)}$ is the $\tau^{(th)}$ largest singular value
10:         $\mathbf{B} \leftarrow \overline{\mathbf{\Sigma}^{(i)}} \mathbf{V}^{(i)^T}$

11: **Procedure** RETURN
12:     **return** $\mathbf{B}$

---

To reduce the algorithm to Misra-Gries, we make the following modifications: each input vector $\mathbf{A}_i$ is an element in $[d]$, and $\mathbf{B}$ is replaced by a dynamic array with a capacity of $m$. The SVD operation is replaced by an aggregation step, where identical elements in $\mathbf{B}$ are grouped together, retaining only one copy of each along with its frequency in $\mathbf{B}$. Consequently, lines 7–10 now correspond to selecting the top-$(\tau - 1)$ elements and reducing their frequencies by $f(\tau)$ [2].

Based on recent work by Liberty (2022), Algorithm 1 possesses the following properties. For completeness, we provide a brief proof in the Appendix.

**Proposition 2.2** ((Liberty, 2022)). *Algorithm 1 uses $O(md)$ space, operates in $O\left( \frac{nm^2 d}{m+1-\tau} \right)$ time, and ensures that $\mathbf{A}^T \mathbf{A} - \mathbf{B}^T \mathbf{B} \succeq 0$. Moreover, it guarantees the following error bound:*

$$\|\mathbf{A}^T \mathbf{A} - \mathbf{B}^T \mathbf{B}\|_2 \le \min_{k \in [0 .. \tau - 1]} \frac{\|\mathbf{A} - [\mathbf{A}]_k\|_F^2}{\tau - k}, \tag{4}$$

*where $\|\cdot\|_2$ is the spectral norm of a matrix, and $[\mathbf{A}]_k$ is the best rank-$k$ approximation of $\mathbf{A}$.*

Note that the error in this context is defined by the maximum distortion rather than a weighted one. If $\tau = (1 - \Omega(1))m$, the running time reduces to $O(nmd)$. Furthermore, for $k = 0$, the error bound simplifies to the original bound established by Liberty (2013). These bounds can be adapted for the Misra-Gries algorithm, where $\mathbf{A}^T \mathbf{A} - \mathbf{B}^T \mathbf{B} \succeq 0$ implies that the algorithm never overestimates element frequencies. Additionally, when implemented with a hash table, the running time for Misra-Gries can be further optimized to $O(n)$.

## 3 LEARNING-AUGMENTED FREQUENT DIRECTION

We aim to augment the Frequent Directions algorithm with learned predictions. The framework is presented in Algorithm 2. Given space for storing $m$ vectors in $\mathbb{R}^d$, the algorithm reserves $m_L \le m$

---

[2]A common implementation of Misra-Gries sets $\tau = m$, and the aggregation step can be optimized using hash tables.

slots for the predicted "frequent directions" $\vec{w}_1, \ldots, \vec{w}_{m_L}$, which are orthonormal vectors returned by a learned oracle. The algorithm then initializes two seperate instances of Algorithm 1, denoted by $\mathcal{A}_{FD}^{\downarrow}$ and $\mathcal{A}_{FD}^{\perp}$, with space usage $m_L$ and $m - 2 \cdot m_L$, respectively.

After initialization, when an input vector $\mathbf{A}_i$ arrives, the algorithm decomposes it into two components, $\mathbf{A}_i = \mathbf{A}_{i,\downarrow} + \mathbf{A}_{i,\perp}$, where $\mathbf{A}_{i,\downarrow}$ is the projection of $\mathbf{A}_i$ onto the subspace spanned by $\vec{w}_1, \ldots, \vec{w}_{m_L}$, and $\mathbf{A}_{i,\perp}$ is the component orthogonal to this subspace. The vector $\mathbf{A}_{i,\downarrow}$ is passed to $\mathcal{A}_{FD}^{\downarrow}$, while $\mathbf{A}_{i,\perp}$ is passed to $\mathcal{A}_{FD}^{\perp}$. Intuitively, $\mathcal{A}_{FD}^{\downarrow}$ is responsible to compute a sketch matrix for the subspace predicted by the learned oracle, whereas $\mathcal{A}_{FD}^{\perp}$ is responsible to compute a sketch matrix for the orthogonal subspace. When the algorithm terminates, the output matrix is obtained by stacking the matrices returned by $\mathcal{A}_{FD}^{\downarrow}$ and $\mathcal{A}_{FD}^{\perp}$. To adapt this framework for the learning-augmented Misra-Gries algorithm, $\mathcal{A}_{FD}^{\downarrow}$ corresponds to an array to record the exact counts of the predicted elements and $\mathcal{A}_{FD}^{\perp}$ corresponds to a Misra-Gries algorithm over all other elements.

---

**Algorithm 2** Learning-Augmented Frequent Direction $\mathcal{A}_{LFD}$

1: **Procedure** INITIALIZATION
2:     **Input:** sketch parameters $m, d \in \mathbb{N}^+$; learned oracle parameter $m_L$ s.t., $m_L \leq m$
3:     Let $\mathbf{P}_H = [\vec{w}_1 \mid \ldots \mid \vec{w}_{m_L}] \in \mathbb{R}^{d \times m_L}$ be the matrix consisting of the $m_L$ orthonormal columns, which are the frequent directions predicted by the learned oracle
4:     Initialize an instance of Algorithm 1: $\mathcal{A}_{FD}^{\downarrow}.initialization(m_L, 0.5 \cdot m_L, d)$
5:     Initialize an instance of Algorithm 1: $\mathcal{A}_{FD}^{\perp}.initialization(m - 2 \cdot m_L, 0.5 \cdot (m - 2 \cdot m_L), d)$

6: **Procedure** UPDATE
7:     **Input:** an input vector $\mathbf{A}_i$
8:     $\mathbf{A}_{i,\downarrow} \leftarrow \mathbf{P}_H \mathbf{P}_H^T \mathbf{A}_i$
9:     $\mathbf{A}_{i,\perp} \leftarrow \mathbf{A}_i - \mathbf{A}_{i,\downarrow}$
10:    $\mathcal{A}_{FD}^{\downarrow}.update(\mathbf{A}_{i,\downarrow})$
11:    $\mathcal{A}_{FD}^{\perp}.update(\mathbf{A}_{i,\perp})$

12: **Procedure** RETURN
13:    $\mathbf{B}^{\downarrow} \leftarrow \mathcal{A}_{FD}^{\downarrow}.return()$
14:    $\mathbf{B}^{\perp} \leftarrow \mathcal{A}_{FD}^{\perp}.return()$
15:    $\mathbf{B} \leftarrow [\mathbf{B}^{\downarrow}; \mathbf{B}^{\perp}]^T$
16:    **return** $\mathbf{B}$

---

### 3.1 THEORETICAL ANALYSIS

We present the theoretical analysis for the (learned) Misra-Gries and (learned) Frequent Directions algorithms under a Zipfian distribution. The error bounds for the (learned) Misra-Gries algorithms are detailed in Theorems 3.1 and 3.2. The corresponding results for the (learned) Frequent Directions algorithm are provided in Theorems 3.3 and 3.4. The complete proofs for are provided in Appendix C and Appendix D, respectively.

Due to space constraints, we provide sketch proofs for the (learned) Misra-Gries algorithm only. The proofs for the (learned) Frequent Directions algorithm follow similar techniques. Since the structure of Misra-Gries is simpler, analyzing its bounds first offers clearer insights into the problems.

**Theorem 3.1** (Expected Error of the Misra-Gries Algorithm). *Given a stream of $n$ elements from a domain $[d]$, where each element $i$ has a frequency $f(i) \propto 1/i$ for $i \in [d]$, the Misra-Gries algorithm using $m$ words of memory achieves expected error of $\mathcal{E}rr \in \Theta\left( \left( \ln \frac{m}{\ln \frac{d}{m}} \right) \cdot \frac{\ln \frac{d}{m}}{(\ln d)^2} \cdot \frac{n}{m} \right)$.* (5)

*Proof Sketch.* At a high level, we first derive an upper bound on the maximum estimation error using Fact 2.2 under the Zipfian distribution assumption. We then partition the elements into two groups: those with frequencies exceeding this error and those that do not. For the first group, the estimation error for each element is bounded by the maximum error. For the second group, since Misra-Gries never overestimates their frequencies, the error is limited to the actual frequency of each element. For each group, we can show that the weighted error is bounded above by the RHS

of (5). For the lower bound, we construct an adversarial input sequence such that the weighted error of elements in the first group indeed matches the upper bound, proving that the bound is tight. □

**Theorem 3.2** (Expected Error of the Learned Misra-Gries Algorithm). *Given a stream of $n$ elements from a domain $[d]$, where each element $i$ has a frequency $f(i) \propto 1/i$ for $i \in [d]$, and assuming a perfect oracle, the learning-augmented Misra-Gries algorithm using $m$ words of memory achieves expected error of $\mathcal{E}rr \in \Theta\left(\frac{1}{m} \cdot \frac{n}{(\ln d)^2}\right)$.*

Here, a perfect oracle is defined as one that makes no mistakes in predicting the top frequent elements. The scenario where the learning oracle is not perfect will be discussed later in this section.

*Proof Sketch.* Under the assumption of access to a perfect oracle, the algorithm does not make estimation error on the top-$m_L$ elements. For the remaining elements, the Misra-Gries algorithm never overestimates its frequency: $\tilde{f}(i) \in [0, f(i)]$. Hence the weighted error is at most

$$\mathcal{E}rr = \sum_{i=m_L+1}^{d} \frac{f(i)}{n} \cdot \left|\tilde{f}(i) - f(i)\right| \leq \sum_{i=m_L+1}^{d} \frac{1}{i \cdot \ln d} \cdot \frac{n}{i \cdot \ln d} \in O\left(\frac{1}{m} \cdot \frac{n}{(\ln d)^2}\right). \quad (6)$$

The lower bound is obtained using a similar technique as in Theorem 3.1, by constructing an input sequence such that the error incurred by the non-predicted elements matches the upper bound. □

**Comparison with Previous Work.** This guarantee matches that of the learning-augmented frequency estimation algorithm of Aamand et al. (2023) but with significant simplifications. Aamand et al. (2023) also reserve separate buckets for the predicted heavy hitters, but to get a robust algorithm in case of faulty predictions, they maintain $O(\log \log n)$ additional CountSketch tables for determining if an arriving element (which is not predicted to be heavy) is in fact a heavy hitter with reasonably high probability. If these tables deem the element light, they output zero as the estimate, and otherwise, they use the estimate of a separate CountSketch table. In contrast, our algorithm uses just a single implementation of the simple and classic Misra-Gries algorithm. This approach has the additional advantage of being deterministic in contrast to CountSketch, which is randomized.

**Robustness and Resilience to Prediction Errors.** We note that the learned Misra-Gries algorithm is *robust* in the sense that it essentially retains the error bounds of its classic counterpart regardless of predictor quality. Indeed, the learned version allocates $m/2$ space to maintain exact counts of elements predicted to be heavy, and uses a classic Misra-Gries sketch of size $m/2$ for the remaining elements. Thus, it incurs no error on the elements predicted to be heavy and on the elements predicted to be light, we get the error guarantees of classic Misra-Gries (using space $m/2$ instead of $m$). It is further worth noting that the error bound of Theorem 3.2 holds even for non-perfect learning oracles or predictions as long as their accuracy is high enough. Specifically, assume that the algorithm allocates some $m_L \in \Omega(m)$ buckets for the learned oracle. Further, assume that only the top $c \cdot m_L$ elements with the highest frequencies are included among the $m_L$ heavy hitters predicted by the oracle, for some $c \leq 1$ (e.g., $c = 0.1$). In this case, Inequality (6) still holds: the summation now starts from $c \cdot m_L + 1$ instead of $m_L + 1$, which does not affect the asymptotic error.

The corresponding theorems for (learned) Frequent Directions are below with proofs in Appendix D.

**Theorem 3.3** (Expected Error of the FREQUENT DIRECTIONS Algorithm). *Assume that the singular values of the input matrix $\mathbf{A}$ to the Algorithm 1 satisfies $\sigma_i^2 \propto \frac{1}{i}$, for all $i \in [d]$, it achieves an expected error of $\mathcal{E}rr(\mathcal{A}_{FD}) \in \Theta\left(\left(\ln \frac{m}{\ln \frac{2d}{m}}\right) \cdot \frac{\ln \frac{d}{m}}{(\ln d)^2} \cdot \frac{\|\mathbf{A}\|_F^2}{m}\right)$.*

**Theorem 3.4** (Expected Error of the Learned FREQUENT DIRECTIONS Algorithm). *Assume that the singular values of the input matrix $\mathbf{A}$ to Algorithm 2 satisfies $\sigma_i^2 \propto \frac{1}{i}$, for all $i \in [d]$, and that learning oracle is perfect, it achieves an expected error of $\mathcal{E}rr(\mathcal{A}_{FD}) \in \Theta\left(\frac{1}{(\ln d)^2} \cdot \frac{\|\mathbf{A}\|_F^2}{m}\right)$.*

**Robustness of Learned Frequent Directions.** It turns out that Algorithm 2 does not come with a robustness guarantee similar to that of Learned Misra-Gries discussed above. In fact, we can construct adversarial inputs for which the expected error is much worse than in the classic setting. Fortunately, there is a way to modify the algorithm slightly using the fact that the residual error $\|\mathbf{A} - [\mathbf{A}]_k\|_F^2$ can be computed within a constant factor using an algorithm from Li et al. (2024).

Since the error of the algorithm scales with the residual error, this essentially allows us to determine if we should output the result of a learned or standard Frequent Directions algorithm. The result is Theorem E.1 on robustness. Combined with Theorem E.2, which explicitly bounds the error of Algorithm 2 in terms of the true and predicted frequent directions, we obtain consistency/robustness tradeoffs for the modified algorithm. Details are provided in Appendix E.

## 4 EXPERIMENTS

We complement our theoretical results with experiments on real data both in the frequency estimation (1-dimensional stream elements) and frequent directions (row vector stream elements) settings. We highlight the main experimental results here and include extensive figures in Appendix F.

### 4.1 FREQUENT DIRECTIONS EXPERIMENTS

**Datasets and Predictions**  We use datasets from Indyk et al. (2019) and Indyk et al. (2021), prior works on learning-based low rank approximation not in the streaming setting. The Hyper dataset (Imamoglu et al., 2018) contains a sequence of hyperspectral images of natural scenes. We consider 80 images each of dimension $1024 \times 768$. The Logo, Friends, and Eagle datasets come from high-resolution Youtube videos[3]. We consider 20 frames from each video each with dimension $3240 \times 1920$. We plot the distribution of singular values for each dataset in Appendix F. For each dataset, we use the top singular vectors of the first matrix in the sequence to form the prediction via a low-rank projection matrix (see Algorithm 2).

**Baselines**  We compare two streaming algorithms and one incomparable baseline. In the streaming setting, we compare the Frequent Directions algorithm of Ghashami et al. (2016) with our learning-augmented variant. Both implementations are based on an existing implementation of Frequent Directions[4]. We additionally plot the performance of the low-rank approximation given by the largest right singular vectors (weighted by singular values). This matrix is *not computable in a stream* as it involves taking the SVD of the entire matrix $A$ but we evaluate it for comparison purposes. Results are displayed based on the rank of the matrix output by the algorithm, which we vary from 20 to 200. For both Frequent Directions and our learned variant, the space used by the streaming algorithm is twice the rank: this is a choice made in the Frequent Directions implementation to avoid running SVD on every insertion and thus improve the update time. We use half of the space for the learned projection component and half for the orthogonal component in our algorithm.

**Results**  For each of the four datasets, we plot tradeoffs between median error (across the sequence of matrices) and rank as well as error across the sequence for a fixed rank of 100 (see Figure 1). We include the latter plots for choices of rank in Appendix F. Our learning-augmented Frequent Directions algorithm improves upon the base Frequent Directions by 1-2 orders of magnitude on all datasets. In most cases, it performs within an order of magnitude of the (full-memory, non-streaming) SVD approximation. In all cases, increasing rank, or equivalently, space, yields significant improvement in the error. These results indicate that learned hints taken from the SVD solution on the first matrix in the sequence can be extremely powerful in improving matrix approximations in streams. As the sequences of matrices retain self-similarity (e.g., due to being a sequence of frames in a video), the predicted projection allows our streaming algorithm to achieve error closer to that of the memory-intensive SVD solution than that of the base streaming algorithm.

### 4.2 FREQUENCY ESTIMATION EXPERIMENTS

**Datasets and Predictions**  We test our algorithm and baselines on the CAIDA (CAIDA, 2016) and AOL (Pass et al., 2006) datasets used in prior work (Hsu et al., 2019; Aamand et al., 2023). The CAIDA dataset contains 50 minutes of internet traffic data, with a stream corresponding to the IP addressed associated with packets passing through an ISP over a minute of data. Each minute of data contains approximately 30 million packets with 1 million unique IPs. The AOL dataset

---

[3]Originally downloaded from `http://youtu.be/L5HQoFIaT4I`, `http://youtu.be/xmLZsEfXEgE` and `http://youtu.be/ufnf_q_3Ofg` and appearing in Indyk et al. (2019).

[4]`https://github.com/edoliberty/frequent-directions`

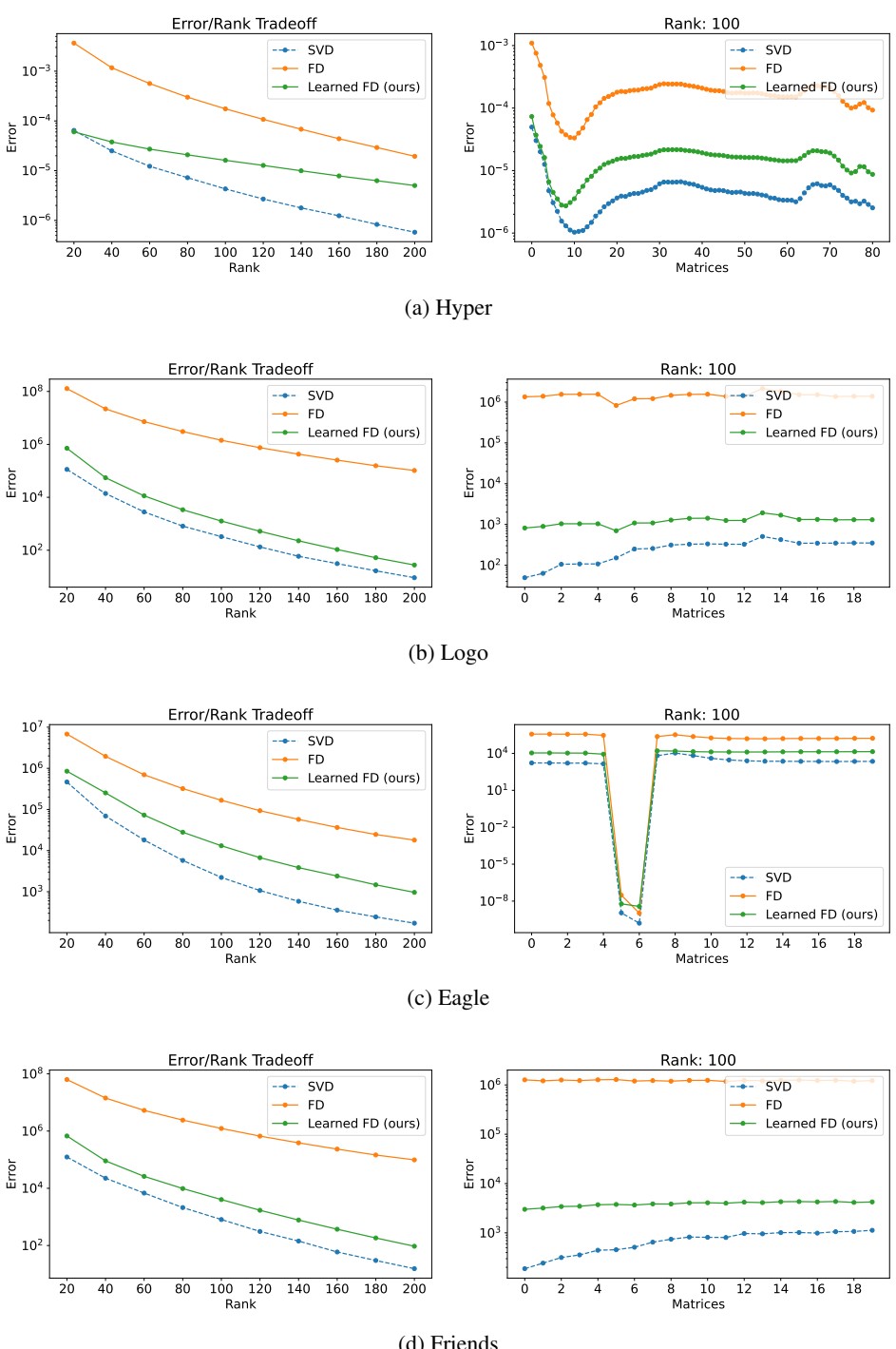

Figure 1: Comparison of matrix approximations. The Frequent Directions and learning-augmented Frequent Directions algorithms are streaming algorithms while the exact SVD stores the entire matrix to compute a low-rank approximation (so it cannot be implemented in a stream). For each dataset, the left plot shows median error (error formula from Equation (2)) as the rank of the approximation varies while the right plot shows error over the sequence of matrices with a fixed rank of 100. The sudden drop in error in Eagle corresponds to several frames of a black screen in the video.

contains 80 days of internet search query data with each stream (corresponding to a day) having around 300k total queries and 100k unique queries. We plot the frequency distribution for both

datasets in Appendix F. We use recurrent neural networks trained in past work of Hsu et al. (2019) as the predictor for both datasets.

**Algorithms**   We compare our learning-augmented Misra-Gries algorithm with learning-augmented CountSketch (Hsu et al., 2019) and learning-augmented CountSketch++ (Aamand et al., 2023). As in Aamand et al. (2023), we forego comparisons against CountMin as it has worse performance both in theory (Aamand et al., 2023) and practice (Hsu et al., 2019). For the prior state-of-the-art, learned CS++, the implemented algorithm does not exactly correspond to the one which achieves the best theoretical bounds as only a single CountSketch table is used (as opposed to two) and the number of rows of the sketch is 3 (as opposed to $O(\log \log n)$). There is a tunable hyperparameter $C$ where elements with estimated frequency less than $Cn/w$ have their estimates truncated to zero (where $n$ is the stream length and $w$ is the sketch width). The space stored by the sketch corresponds to $3w$ as there are 3 rows. For Misra-Gries, the space corresponds to double the number of stored counters as each counter requires storing a key as well as a count. As in prior work, for all algorithms, their learned variants use half of the space for the normal algorithm and half of the space to store exact counts for the elements with top predicted frequencies.

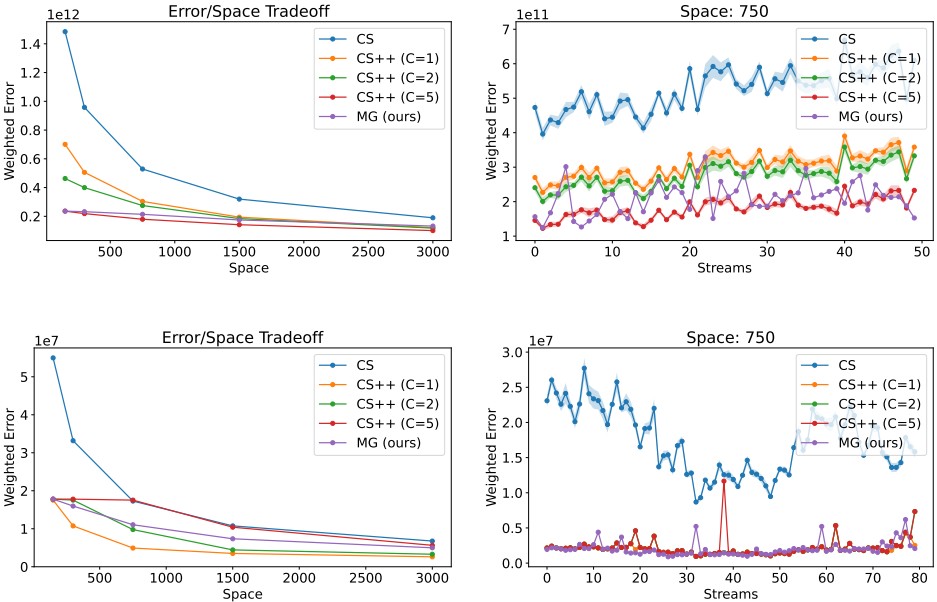

Figure 2: Comparison of learning-augmented frequency estimation algorithms. Top: CAIDA, Bottom: AOL. For both datasets, the left plot show the median error of each method (across all 50 streams) with varying space budgets. The right plot shows the performance of each algorithm across streams with fixed space of 750 words. Randomized algorithms are averaged across 10 trials and one standard deviation is shaded.

**Results**   For both datasets, we compare the learning-augmented algorithms by plotting the tradeoff between median error and space as well as error across the sequence of streams for a fixed space of 750 (see Figure 2). In Appendix F, we include the latter plots for all choices of space, as well as all corresponding plots both without predictions (to compare the base CS, CS++, and MG algorithms) and under unweighted error (taken as the unweighted sum of absolute errors over all stream items regardless of frequency) which was also evaluated in prior work. The learning-augmented Misra-Gries algorithm improves significantly over learning-augmented CountSketch, as implied by our theoretical bounds. Furthermore, it is competitive with the state-of-the-art learning-augmented CS++ algorithm. Sometimes our algorithm outperforms the best hyperparameter choice CS++ and often outperforms several of the hyperparameter choices of CS++. Furthermore, learning-augmented MG has no equivalent tunable parameter and is simpler to deploy (especially as CS++ is already a simplification of the theoretical algorithm of Aamand et al. (2023)). As learning-augmented MG is the only *deterministic* algorithm with provable guarantees in the setting of Hsu et al. (2019), our results indicate that there is essentially no cost to derandomization.

ACKNOWLEDGMENTS

Justin Y. Chen is supported by an NSF Graduate Research Fellowship under Grant No. 17453. Hao WU was a Postdoctoral Fellow at the University of Copenhagen, supported by Providentia, a Data Science Distinguished Investigator grant from Novo Nordisk Fonden. Siddharth Gollapudi is supported in part by the NSF (CSGrad4US award no. 2313998).

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

# Appendix

## Table of Contents

## Table of Notations

| Symbol | Definition |
|---|---|
| $n$ | Number of Inputs in the Stream |
| $d$ | Domain Size (Frequency Estimation)
Dimension (Frequent Directions) |
| $f(\cdot)$ | Frequency |
| $a_i$ | The $i^{\text{th}}$ input element in the stream (Frequency Estimation) |
| $\mathbf{A}_i$ | The $i^{\text{th}}$ input vector in the stream (Frequent Directions) |
| $\mathbf{A}$ | Stream Input Matrix |
| $\vec{e}_i$ | Standard Basis Vector |
| $\sigma_i$ | Singular Value of a Matrix |
| $N(\cdot, \cdot)$ | Normal Distribution |
| $\mathbf{U}^{(i)}, \mathbf{\Sigma}^{(i)}, \mathbf{V}^{(i)}$ | The SVD decomposition matrices at the $i^{\text{th}}$ iteration of Algorithm 1 |

Table 3: Definitions of Main Notation.

# A    OTHER RELATED WORKS

There is also a vast literature on sketching based algorithms for low-rank approximation without any learned-augmentation (Sarlos, 2006; Ghashami & Phillips, 2014b; Liberty, 2013; Tropp et al., 2017; Meyer et al., 2021). We refer to the monograph Woodruff et al. (2014) for more details.

# B    MISSING PROOFS FOR PRELIMINARIES

We refer the reader to Section 2 for the full statements.

*Proof of Lemma 2.1.*

$$
\begin{aligned}
\mathop{\mathbb{E}}_{\vec{v} \sim N(0, \mathbf{A}^T \mathbf{A})} \left[ \|\mathbf{A}\vec{v}\|_2^2 - \tilde{f}(\vec{v}) \right] &= \mathbb{E}\left[\vec{v}^T \mathbf{A}^T \mathbf{A} \vec{v}\right] - \mathbb{E}\left[\vec{v}^T \mathbf{B}^T \mathbf{B} \vec{v}\right] \\
&= \mathbb{E}\left[\text{tr}(\vec{v}^T \mathbf{A}^T \mathbf{A} \vec{v})\right] - \mathbb{E}\left[\text{tr}(\vec{v}^T \mathbf{B}^T \mathbf{B} \vec{v})\right] \\
&= \mathbb{E}\left[\text{tr}(\mathbf{A}^T \mathbf{A} x x^T)\right] - \mathbb{E}\left[\text{tr}(\mathbf{B}^T \mathbf{B} x x^T)\right] \\
&= \text{tr}(\mathbf{A}^T \mathbf{A}\, \mathbb{E}\left[x x^T\right]) - \text{tr}(\mathbf{B}^T \mathbf{B}\, \mathbb{E}\left[x x^T\right]) \\
&= \text{tr}((\mathbf{A}^T \mathbf{A})^2) - \text{tr}(\mathbf{B}^T \mathbf{B} \mathbf{A}^T \mathbf{A})
\end{aligned}
$$

Let $\vec{v}_i$ be the right singular vectors of $\mathbf{A}$:

$$
\text{tr}\left(\left(\mathbf{A}^T \mathbf{A}\right)^2\right) = \text{tr}\left(\left(\sum_{i \in [d]} \sigma_i^2 \cdot \vec{v}_i \vec{v}_i^T\right)^2\right) \tag{7}
$$

$$
= \text{tr}\left(\sum_{i \in [d]} \sigma_i^4 \cdot \vec{v}_i \vec{v}_i^T\right) = \sum_{i \in [d]} \sigma_i^4, \tag{8}
$$

$$
\text{tr}\left(\mathbf{B}^T \mathbf{B} \mathbf{A}^T \mathbf{A}\right) = \text{tr}\left(\mathbf{B}^T \mathbf{B}\left(\sum_{i \in [d]} \sigma_i^2 \cdot \vec{v}_i \vec{v}_i^T\right)\right) \tag{9}
$$

$$
= \sum_{i \in [d]} \sigma_i^2 \cdot \text{tr}\left(\mathbf{B}^T \mathbf{B} \vec{v}_i \vec{v}_i^T\right) = \sum_{i \in [d]} \sigma_i^2 \cdot \text{tr}\left(\vec{v}_i^T \mathbf{B}^T \mathbf{B} \vec{v}_i\right). \tag{10}
$$

Therefore,

$$
\mathop{\mathbb{E}}_{\vec{v} \sim N(0, \mathbf{A}^T \mathbf{A})} \left[\vec{v}^T (\mathbf{A}^T \mathbf{A} - \mathbf{B}^T \mathbf{B}) \vec{v}\right] = \sum_{i \in [d]} \sigma_i^4 - \sum_{i \in [d]} \sigma_i^2 \cdot \text{tr}\left(\vec{v}_i^T \mathbf{B}^T \mathbf{B} \vec{v}_i\right). \tag{11}
$$

Further, since $\tilde{f}(\vec{v}) = \|\mathbf{B}\vec{v}\|_2^2$ and $0 \leq \tilde{f}(\vec{v}) \leq f(\vec{v})$, Equation (2) can be written as

$$
\sum_{i \in [d]} \frac{\sigma_i^2}{\|\mathbf{A}\|_F^2} \cdot \vec{v}_i^T (\mathbf{A}^T \mathbf{A} - \mathbf{B}^T \mathbf{B}) \vec{v}_i. \tag{12}
$$

The first term is given by

$$
\sum_{i \in [d]} \sigma_i^2 \cdot \vec{v}_i^T \left(\sum_{j \in [d]} \sigma_j^2 \vec{v}_j \vec{v}_j^T\right) \vec{v}_i = \sum_{i \in [d]} \sigma_i^4 \cdot \vec{v}_i^T \vec{v}_i = \sum_{i \in [d]} \sigma_i^4. \tag{13}
$$

Further note that

$$
\mathop{\mathbb{E}}_{\vec{v} \sim N(0, \mathbf{A}^T \mathbf{A})} \left[\|\vec{v}\|_2^2\right] = \mathop{\mathbb{E}}_{z \sim N(0, I)} \left[z^T \mathbf{A} \mathbf{A}^T z\right] = \mathop{\mathbb{E}}_{z \sim N(0, I)} \left[\text{tr}\left(z^T \mathbf{A} \mathbf{A}^T z\right)\right] \tag{14}
$$

$$
= \mathop{\mathbb{E}}_{z \sim N(0, I)} \left[\text{tr}\left(\mathbf{A} \mathbf{A}^T z z^T\right)\right] = \text{tr}\left(\mathbf{A} \mathbf{A}^T\right) = \|\mathbf{A}\|_F^2. \tag{15}
$$

Therefore,

$$\sum_{i \in [d]} \frac{\sigma_i^2}{\|\mathbf{A}\|_F^2} \cdot \vec{v}_i^T (\mathbf{A}^T \mathbf{A} - \mathbf{B}^T \mathbf{B}) \vec{v}_i = \frac{\mathbb{E}_{\vec{v} \sim N(0, \mathbf{A}^T \mathbf{A})} \left[ \vec{v}^T (\mathbf{A}^T \mathbf{A} - \mathbf{B}^T \mathbf{B}) \vec{v} \right]}{\mathbb{E}_{\vec{v} \sim N(0, \mathbf{A}^T \mathbf{A})} \left[ \|\vec{v}\|_2^2 \right]}. \tag{16}$$

$\square$

*Proof of Fact 2.2.* For consistence, at each iteration, if $\mathbf{B}$ has less than $m$ rows after insertion, we still define

$$\mathbf{U}^{(i)}, \mathbf{\Sigma}^{(i)}, \mathbf{V}^{(i)} \leftarrow \text{SVD}(\mathbf{B}), \text{ and } \overline{\mathbf{\Sigma}^{(i)}} = \mathbf{\Sigma}^{(i)}.$$

To analyze the error, let $\mathbf{B}^{(i)}$ denote the value of $\mathbf{B}$ after the $i^{(th)}$ iteration, and define

$$\begin{aligned}
\mathbf{\Delta}^{(i)} &\doteq \mathbf{A}_i^T \mathbf{A}_i + \mathbf{B}^{(i-1)^T} \mathbf{B}^{(i-1)} - \mathbf{B}^{(i)^T} \mathbf{B}^{(i)} \\
&= \mathbf{V}^{(i)} \mathbf{\Sigma}^{(i)^T} \mathbf{U}^{(i)^T} \mathbf{U}^{(i)} \mathbf{\Sigma}^{(i)} \mathbf{V}^{(i)^T} - \mathbf{V}^{(i)} \overline{\mathbf{\Sigma}^{(i)}}^T \overline{\mathbf{\Sigma}^{(i)}} \mathbf{V}^{(i)^T} \\
&= \mathbf{V}^{(i)} \left( \mathbf{\Sigma}^{(i)^T} \mathbf{\Sigma}^{(i)} - \overline{\mathbf{\Sigma}^{(i)}}^T \overline{\mathbf{\Sigma}^{(i)}} \right) \mathbf{V}^{(i)^T}.
\end{aligned}$$

Then

$$\mathbf{A}^T \mathbf{A} - \mathbf{B}^{(n)^T} \mathbf{B}^{(n)} = \sum_{i \in [n]} \left( \mathbf{A}_i^T \mathbf{A}_i + \mathbf{B}^{(i-1)^T} \mathbf{B}^{(i-1)} - \mathbf{B}^{(i)^T} \mathbf{B}^{(i)} \right) = \sum_{i \in [n]} \mathbf{\Delta}^{(i)}. \tag{17}$$

Since each $\mathbf{\Delta}^{(i)} \succeq 0$, we prove that

$$\mathbf{A}^T \mathbf{A} - \mathbf{B}^{(n)^T} \mathbf{B}^{(n)} \succeq 0. \tag{18}$$

Let $\vec{v}_1, \ldots, \vec{v}_d \in \mathbb{R}^k$ be the right singular vectors of $\mathbf{A}$. For each $k = 0, \ldots, \tau - 1$, define the projection matrix $\overline{\mathbf{P}_k} = [\vec{0} \mid \ldots \mid \vec{0} \mid \vec{v}_{k+1} \mid \ldots \mid \vec{v}_d] \in \mathbb{R}^{d \times d}$, consisting of columns vectors $\vec{0}, \ldots, \vec{0}, \vec{v}_{k+1}, \ldots, \vec{v}_d$. The null space is thus spanned by the top-$k$ right singular vectors of $\mathbf{A}$. We claim the following holds:

$$\left\| \mathbf{\Delta}^{(i)} \right\|_2 \leq \frac{1}{\tau - k} \cdot \text{tr} \left( \overline{\mathbf{P}_k}^T \mathbf{\Delta}^{(i)} \overline{\mathbf{P}_k} \right), \qquad \forall k = 0, \ldots, \tau - 1. \tag{19}$$

Before proving it, we complete the proof of the error:

$$\left\| \mathbf{A}^T \mathbf{A} - \mathbf{B}^{(n)^T} \mathbf{B}^{(n)} \right\|_2 = \left\| \sum_{i \in [n]} \mathbf{\Delta}^{(i)} \right\|_2 \leq \sum_{i \in [n]} \left\| \mathbf{\Delta}^{(i)} \right\|_2 \leq \frac{1}{\tau - k} \cdot \sum_{i \in [n]} \text{tr} \left( \overline{\mathbf{P}_k}^T \mathbf{\Delta}^{(i)} \overline{\mathbf{P}_k} \right) \tag{20}$$

$$= \frac{1}{\tau - k} \cdot \text{tr} \left( \overline{\mathbf{P}_k}^T \left( \sum_{i \in [n]} \mathbf{\Delta}^{(i)} \right) \overline{\mathbf{P}_k} \right) \tag{21}$$

$$= \frac{1}{\tau - k} \cdot \text{tr} \left( \overline{\mathbf{P}_k}^T \mathbf{A}^T \mathbf{A} \overline{\mathbf{P}_k} \right) = \frac{1}{\tau - k} \cdot \| \mathbf{A} - [\mathbf{A}]_k \|_F^2. \tag{22}$$

*Proof of Inequality* (19): First,

$$\begin{aligned}
\left\| \mathbf{\Delta}^{(i)} \right\|_2 &= \left\| \mathbf{V}^{(i)} \left( \mathbf{\Sigma}^{(i)^T} \mathbf{\Sigma}^{(i)} - \overline{\mathbf{\Sigma}^{(i)}}^T \overline{\mathbf{\Sigma}^{(i)}} \right) \mathbf{V}^{(i)^T} \right\|_2 \\
&= \left\| \mathbf{\Sigma}^{(i)^T} \mathbf{\Sigma}^{(i)} - \overline{\mathbf{\Sigma}^{(i)}}^T \overline{\mathbf{\Sigma}^{(i)}} \right\|_2 \\
&= \left\| \min\{ \mathbf{\Sigma}^{(i)^2}, \left( \sigma_\tau^{(i)} \right)^2 \mathbf{I} \} \right\|_2 = \left( \sigma_\tau^{(i)} \right)^2,
\end{aligned}$$

where $\sigma_\tau^{(i)}$ is the $\tau^{(th)}$ largest singular value of $\mathbf{\Sigma}^{(i)}$. Next,

$$\text{tr}\left(\mathbf{\Delta}^{(i)}\right) = \text{tr}\left(\mathbf{V}^{(i)}\left(\mathbf{\Sigma}^{(i)T}\mathbf{\Sigma}^{(i)} - \overline{\mathbf{\Sigma}^{(i)}}^T\overline{\mathbf{\Sigma}^{(i)}}\right)\mathbf{V}^{(i)T}\right) \tag{23}$$

$$= \text{tr}\left(\mathbf{\Sigma}^{(i)T}\mathbf{\Sigma}^{(i)} - \overline{\mathbf{\Sigma}^{(i)}}^T\overline{\mathbf{\Sigma}^{(i)}}\right) = \sum_{j\in[d]}\left(\sigma_j^{(i)}\right)^2 \geq \tau \cdot \left(\sigma_\tau^{(i)}\right)^2. \tag{24}$$

Next, let $\boldsymbol{P}_k = [\vec{v}_1 \mid \ldots \mid \vec{v}_k \mid \vec{0} \mid \ldots \mid \vec{0}] \in \mathbb{R}^{d\times d}$ be the projection matrix to the space spanned by the top-$k$ right singular vectors of $\mathbf{A}$. Then $\boldsymbol{P}_k + \overline{\boldsymbol{P}_k} = \boldsymbol{I}$, and

$$\text{tr}\left(\mathbf{\Delta}^{(i)}\right) = \text{tr}\left(\left(\boldsymbol{P}_k + \overline{\boldsymbol{P}_k}\right)^T\mathbf{\Delta}^{(i)}\left(\boldsymbol{P}_k + \overline{\boldsymbol{P}_k}\right)\right) = \text{tr}\left(\boldsymbol{P}_k^T\mathbf{\Delta}^{(i)}\boldsymbol{P}_k\right) + \text{tr}\left(\overline{\boldsymbol{P}_k}^T\mathbf{\Delta}^{(i)}\overline{\boldsymbol{P}_k}\right). \tag{25}$$

Expanding $\text{tr}\left(\boldsymbol{P}_k^T\mathbf{\Delta}^{(i)}\boldsymbol{P}_k\right)$ we get

$$\text{tr}\left(\boldsymbol{P}_k^T\mathbf{\Delta}^{(i)}\boldsymbol{P}_k\right) = \sum_{j\in[k]}\vec{v}_j^T\mathbf{\Delta}^{(i)}\vec{v}_j \leq k \cdot \left(\sigma_\tau^{(i)}\right)^2, \tag{26}$$

which implies that

$$\text{tr}\left(\overline{\boldsymbol{P}_k}^T\mathbf{\Delta}^{(i)}\overline{\boldsymbol{P}_k}\right) \leq (\tau - k) \cdot \left(\sigma_\tau^{(i)}\right)^2 = (\tau - k) \cdot \left\|\mathbf{\Delta}^{(i)}\right\|_2^2. \tag{27}$$

**Running Time:** For each $\mathbf{A}_i$, inserting it into $\mathbf{B}$ takes $O(d)$ time. When $\mathbf{B}$ reaches its capacity of $m$ rows, the operations in Lines 7-10 are triggered, and performing the SVD requires $O(m^2 d)$ time.

After completing this step, $\mathbf{B}$ has at least $m - \tau + 1$ empty rows. Thus, the algorithm can accommodate at least $m - \tau + 1$ additional insertions before Lines 7-10 need to be executed again. Consequently, the total running time is:

$$O\left(\frac{n}{m - \tau + 1} \cdot m^2 d\right). \tag{28}$$

$\square$

## C   ANALYSIS OF MISRA-GRIES

**Proof of Theorem 3.1.** We need to prove both upper bounds and lower bounds for the theorem.

*Upper Bound for Theorem 3.1.*   W.L.O.G., assume that $f(1) \geq f(2) \geq \cdots \geq f(d)$. Since $f(i) \propto \frac{1}{i}$ for each $i \in [d]$, and the input stream consists of $n$ elements, it follows that $f(i) \approx \frac{n}{i \cdot \ln d}$.

Assume that we have a Misra-Gries sketch of size $m \in \mathbb{N}^+$. Then by translating the error guarantee from Fact 2.2 for Misra-Gries, we have

$$\max_{i \in [d]} \left| \tilde{f}(i) - f(i) \right| \leq \min_{k \in [0 \,..\, m-1]} \frac{n - \sum_{j \in [k]} f(i)}{\tau - k} \leq \frac{n - \sum_{j \in [m/2]} f(i)}{m/2} = \frac{2 \cdot n \cdot \ln \frac{2d}{m}}{m \cdot \ln d}. \quad (29)$$

Further,

$$i \geq \frac{m}{2 \cdot \ln \frac{2d}{m}} \implies f(i) = \frac{n}{i \cdot \ln d} \leq \frac{2 \cdot n \cdot \ln \frac{2d}{m}}{m \cdot \ln d} \quad (30)$$

Since we also know that $0 \leq \tilde{f}(i) \leq f(i)$, it follows that

$$\sum_{i \in [d} \frac{f(i)}{n} \cdot \left| \tilde{f}(i) - f(i) \right| = \sum_{i=1}^{\frac{m}{2 \cdot \ln \frac{2d}{m}}} \frac{f(i)}{n} \cdot \left| \tilde{f}(i) - f(i) \right| + \sum_{i = \frac{m}{2 \cdot \ln \frac{2d}{m}} + 1}^{d} \frac{f(i)}{n} \cdot \left| \tilde{f}(i) - f(i) \right|$$

$$\leq \sum_{i=1}^{\frac{m}{2 \cdot \ln \frac{2d}{m}}} \frac{f(i)}{n} \cdot \frac{2 \cdot n \cdot \ln \frac{2d}{m}}{m \cdot \ln d} + \sum_{i = \frac{m}{2 \cdot \ln \frac{2d}{m}} + 1}^{d} \frac{f(i)}{n} \cdot f(i)$$

$$\leq \sum_{i=1}^{\frac{m}{2 \cdot \ln \frac{2d}{m}}} \frac{1}{i \cdot \ln d} \cdot \frac{2 \cdot n \cdot \ln \frac{2d}{m}}{m \cdot \ln d} + \sum_{i = \frac{m}{2 \cdot \ln \frac{2d}{m}} + 1}^{d} \frac{1}{i \cdot \ln d} \cdot \frac{n}{i \cdot \ln d}$$

$$\in O \left( \frac{\ln \frac{m}{2 \cdot \ln \frac{2d}{m}}}{\ln d} \cdot \frac{n \cdot \ln \frac{d}{m}}{m \cdot \ln d} + \frac{1}{\frac{m}{2 \cdot \ln \frac{2d}{m}} + 1} \cdot \frac{n}{(\ln d)^2} \right)$$

$$= O \left( \frac{\ln \frac{m}{2 \cdot \ln \frac{2d}{m}}}{1} \cdot \frac{n \cdot \ln \frac{d}{m}}{m \cdot (\ln d)^2} + \frac{\ln \frac{d}{m}}{m} \cdot \frac{n}{(\ln d)^2} \right)$$

$$= O \left( \left( \ln \frac{m}{\ln \frac{2d}{m}} \right) \cdot \frac{\ln \frac{d}{m}}{m} \cdot \frac{n}{(\ln d)^2} \right)$$

*Lower Bound for Theorem 3.1.*   To prove the lower bound, we assume there is an adversary which controls the order that the input elements arrive, under the constraints that $\sum_{i \in [d]} f(i) = n$ and $f(i) \propto 1/i$, to maximize the error of the Misra-Gries algorithm.

Denote $\mathbf{B}$ the array maintained by the Misra-Gries algorithm, containing $m$ buckets. Initially, all buckets are empty.

First, the adversary inserts elements $1, \ldots, t$ to the Misra-Gries algorithm, with multiplicities $f(1), \ldots, f(t)$, where $t = \frac{m}{\ln \frac{2d}{m}}$. After this, $\mathbf{B}$ contains $t$ non-empty buckets (for simplicity, here we assume that $\sum_{i \in [t]} f(i)$ is a multiple of $m$), which stores elements $1, \ldots, t$, associated with their recorded frequencies $f(1), \ldots, f(t)$, which we call their counters.

Next, let $\mathcal{C}$ be the multi-set consisting of elements $t+1, \ldots, d$, such that each element $i \in [t+1 \,..\, d]$ has multiplicity $f(i)$ in $\mathcal{C}$. Consider the following game:

- Adversary: pick an element $i$ from $\mathcal{C}$ that is not in $\mathbf{B}$. If such element exists, remove one copy of it from $\mathcal{C}$, and send it to the Misra-Gries algorithm as the next input. If there is no such element, stop the game.

- Misra-Gries Algorithm: process the input $i$.

During this game, $\mathbf{B}$ are filled up and contains no empty bucket after at most every $m$ input, and the counters of elements $1, \ldots, t$ decrease by 1 (if they are still above zero) when $\mathbf{B}$ is updated to make empty buckets.

Further, when the game stops, there can be at most $m$ distinct elements in $\mathcal{C}$, with frequency sum at most $\sum_{i=t+1}^{t+m} f(i)$. It follows that the counters of elements $1, \ldots, t$ in $\mathbf{B}$ decrease at least by

$$\frac{1}{m} \cdot \sum_{i=m+t+1}^{d} f(i) \in \Omega\left(\frac{n \cdot \ln \frac{d}{m}}{m \cdot \ln d}\right), \quad \text{since } m + t + 1 \leq 2m.$$

Therefore, the weighted error introduced by these counters is at least

$$\Omega\left(\sum_{i \in [t]} \frac{1}{i \cdot \ln d} \cdot \frac{n \cdot \ln \frac{d}{m}}{m \cdot \ln d}\right) = \Omega\left(\frac{\ln t}{\ln d} \cdot \frac{n \cdot \ln \frac{d}{m}}{m \cdot \ln d}\right) = \Omega\left(\left(\ln \frac{m}{\ln \frac{2d}{m}}\right) \cdot \frac{\ln \frac{d}{m}}{(\ln d)^2} \cdot \frac{n}{m}\right).$$

□

**Proof of Theorem 3.2.** We need to prove both upper bounds and lower bounds for the theorem.

*Upper Bound for Theorem 3.2.* It suffices to show that, there exists a parameter setting of $m_L$ which enables the algorithm to achieve the desired error bound.

Assume that the algorithm reserves $m_L = m/3$ words for the learned oracle. Then for each element $i \in [m_L]$, its frequency estimate $\tilde{f}(i) = f(i)$. And for each $i \notin [m_L]$, the Misra-Gries algorithm never overestimate its frequency: $\tilde{f}(i) \in [0, f(i)]$. Hence

$$\mathcal{E}rr = \sum_{i=m_L+1}^{d} \frac{f(i)}{n} \cdot \left|\tilde{f}(i) - f(i)\right| \leq \sum_{i=m_L+1}^{d} \frac{1}{i \cdot \ln d} \cdot \frac{n}{i \cdot \ln d} \in O\left(\frac{1}{m} \cdot \frac{n}{(\ln d)^2}\right). \tag{31}$$

*Lower Bound for Theorem 3.2.* To establish the lower bound, we consider an adversarial scenario where an adversary controls the order in which elements arrive, subject to the constraints $\sum_{i \in [d]} f(i) = n$ and $f(i) \propto 1/i$. The adversary's goal is to maximize the error of the learned Misra-Gries algorithm.

According to the framework presented in Algorithm 2 for the learned Frequent Directions, the learned Misra-Gries algorithm initializes two separate Misra-Gries instances: one for the $m_L$ elements predicted to be frequent and one for elements predicted to be non-frequent.

Since $m_L$ memory words are already reserved for storing the frequencies of the predicted frequent elements, we do not need to run a full Misra-Gries algorithm on the these elements. Instead, we only record their observed frequencies.

By overloading the notation a little, let us denote $\mathbf{B}$ as the array used by the Misra-Gries instance managing the predicted non-frequent elements, which has a capacity of $m - m_L$ buckets. Initially, all buckets in $\mathbf{B}$ are empty.

Since the learned Misra-Gries algorithm incurs no estimation error for the predicted frequent elements, our analysis focuses on the non-frequent elements and the potential error introduced by the Misra-Gries instance that processes them.

Let $\mathcal{C}$ denote the multi-set of elements $m_L + 1, \ldots, d$, where each element $i \in [m_L + 1 .. d]$ appears with multiplicity $f(i)$ in $\mathcal{C}$. Consider the following adversarial game:

- Adversary's Role: At each step, the adversary selects an element $i$ from $\mathcal{C}$ that is not currently stored in the array $\mathbf{B}$. If such an element exists, the adversary removes one occurrence of $i$ from $\mathcal{C}$ and sends it to the Misra-Gries algorithm as the next input. If there is no such element left, the adversary halts the game.

- Misra-Gries Algorithm's Role: The Misra-Gries algorithm processes the incoming element $i$ as it would normally, using the array $\mathbf{B}$ of capacity $m - m_L$.

After the game, the remaining elements in $\mathcal{C}$ are fed to the Misra-Gries algorithm in arbitrary order by the adversary.

Now, consider the estimation error made by the Misra-Gries algorithm on the elements $m_L + 1, \ldots, m_L + 2(m - m_L)$. Since the array $\mathbf{B}$ can only store up to $m - m_L$ elements, the algorithm must estimate the frequency of at least $m - m_L$ elements from this range as zero. Therefore, the error is at least

$$\sum_{i=m_L+(m-m_L)+1}^{m_L+2(m-m_L)} \frac{f(i)}{n} \cdot f(i) = \sum_{i=m-m_L+1}^{m_L+2(m-m_L)} \frac{1}{i \cdot \ln d} \cdot \frac{n}{i \cdot \ln d} \in \Omega\left(\frac{n}{m \cdot \ln^2 d}\right),$$

which finishes the proof.

$\square$

# D  ANALYSIS OF FREQUENT DIRECTIONS

## D.1  FREQUENT DIRECTIONS UNDER ZIPFIAN

**Proof of Theorem 3.3.**  We need to prove both upper bounds and lower bounds for the theorem.

*Upper Bound for Theorem 3.3.*  First, based on the assumption that $\sigma_i^2 \propto \frac{1}{i}$, and Fact 2.2, we have

$$\|\mathbf{A}^T\mathbf{A} - \mathbf{B}^T\mathbf{B}\|_2 \leq \min_{k \in [0 \, . \, . \, m-1]} \frac{\|\mathbf{A} - [\mathbf{A}]_k\|_F^2}{\tau - k} \tag{32}$$

$$= \min_{k \in [0 \, . \, . \, m-1]} \frac{\sum_{i=k+1}^d \sigma_i^2}{\tau - k} \tag{33}$$

$$= \min_{k \in [0 \, . \, . \, m-1]} \frac{(H_d - H_k) \cdot \|\mathbf{A}\|_F^2}{(\tau - k) \cdot \ln d} \tag{34}$$

$$\leq \frac{2 \cdot (H_d - H_{m/2}) \cdot \|\mathbf{A}\|_F^2}{m \cdot \ln d} \tag{35}$$

$$\in O\left(\frac{\|\mathbf{A}\|_F^2 \cdot \ln \frac{2d}{m}}{m \cdot \ln d}\right), \tag{36}$$

where $H_m \doteq \sum_{j \in [m]} \frac{1}{j}, \forall m \in \mathbb{N}^+$ are the harmonic numbers. Further,

$$i \geq \frac{m}{\ln \frac{2d}{m}} \implies \sigma_i^2 = \frac{\|\mathbf{A}\|_F^2}{i \cdot \ln d} \leq \frac{\|\mathbf{A}\|_F^2 \cdot \ln \frac{2d}{m}}{m \cdot \ln d} \tag{37}$$

Since $\mathbf{B}^T\mathbf{B} \succeq 0$, and $\mathbf{A}^T\mathbf{A} - \mathbf{B}^T\mathbf{B} \succeq 0$ by Fact 2.2, it follows that for each right singular vector $\vec{v}_i$ of $\mathbf{A}$

$$0 \leq \vec{v}_i^T(\mathbf{A}^T\mathbf{A} - \mathbf{B}^T\mathbf{B})\vec{v}_i \leq \vec{v}_i^T\mathbf{A}^T\mathbf{A}\vec{v}_i \leq \sigma_i^2, \tag{38}$$

where $\sigma_i$ is the singular value associated with $\vec{v}_i$.

Therefore, the expected error is given by

$$\mathcal{E}rr(\mathcal{A}_{FD}) \doteq \sum_{i \in [d]} \frac{\sigma_i^2}{\|\mathbf{A}\|_F^2} \cdot \vec{v}_i^T(\mathbf{A}^T\mathbf{A} - \mathbf{B}^T\mathbf{B})\vec{v}_i \tag{39}$$

$$= \sum_{i=1}^{\frac{m}{\ln \frac{2d}{m}}} \frac{\sigma_i^2}{\|\mathbf{A}\|_F^2} \cdot \vec{v}_i^T(\mathbf{A}^T\mathbf{A} - \mathbf{B}^T\mathbf{B})\vec{v}_i + \sum_{i=\frac{m}{\ln \frac{2d}{m}}+1}^{d} \frac{\sigma_i^2}{\|\mathbf{A}\|_F^2} \cdot \vec{v}_i^T(\mathbf{A}^T\mathbf{A} - \mathbf{B}^T\mathbf{B})\vec{v}_i \tag{40}$$

$$\in O\left(\sum_{i=1}^{\frac{m}{\ln \frac{2d}{m}}} \frac{\sigma_i^2}{\|\mathbf{A}\|_F^2} \cdot \frac{\|\mathbf{A}\|_F^2 \cdot \ln \frac{2d}{m}}{m \cdot \ln d} + \sum_{i=\frac{m}{\ln \frac{2d}{m}}+1}^{d} \frac{\sigma_i^2}{\|\mathbf{A}\|_F^2} \cdot \sigma_i^2\right) \tag{41}$$

$$= O\left(\sum_{i=1}^{\frac{m}{\ln \frac{2d}{m}}} \frac{1}{i \cdot \ln d} \cdot \frac{\|\mathbf{A}\|_F^2 \cdot \ln \frac{2d}{m}}{m \cdot \ln d} + \sum_{i=\frac{m}{\ln \frac{2d}{m}}+1}^{d} \frac{1}{i \cdot \ln d} \cdot \frac{\|\mathbf{A}\|_F^2}{i \cdot \ln d}\right) \tag{42}$$

$$= O\left(\frac{\ln \frac{m}{\ln \frac{2d}{m}}}{\ln d} \cdot \frac{\|\mathbf{A}\|_F^2 \cdot \ln \frac{2d}{m}}{m \cdot \ln d} + \frac{1}{\frac{m}{\ln \frac{2d}{m}}+1} \cdot \frac{\|\mathbf{A}\|_F^2}{(\ln d)^2}\right) \tag{43}$$

$$= O\left(\frac{\ln \frac{m}{\ln \frac{2d}{m}}}{1} \cdot \frac{\|\mathbf{A}\|_F^2 \cdot \ln \frac{2d}{m}}{m \cdot (\ln d)^2} + \frac{\ln \frac{2d}{m}}{m} \cdot \frac{\|\mathbf{A}\|_F^2}{(\ln d)^2}\right) \tag{44}$$

$$= O\left(\left(\ln \frac{m}{\ln \frac{2d}{m}}\right) \cdot \frac{\ln \frac{d}{m}}{m} \cdot \frac{\|\mathbf{A}\|_F^2}{(\ln d)^2}\right) \tag{45}$$

*Lower Bound for Theorem 3.3.* The proof of the lower bound follows the same approach as the one for Theorem 3.1, in Appendix C.

Assume that $\mathbf{A}$ consists of standard basis vectors $\vec{e}_1, \dots, \vec{e}_d \in \mathbb{R}^d$. Let $f(\vec{e}_i)$ denote the number of occurrences of $\vec{e}_i$ in $\mathbf{A}$. Without loss of generality, assume that $f(\vec{e}_1) \geq \dots \geq f(\vec{e}_d)$. In this case, we have $f(\vec{e}_i) = \sigma_i^2$ and $\sum_{i \in [d]} f(\vec{e}_i) = \sum_{i \in [d]} \sigma_i^2 = \|\mathbf{A}\|_F^2 = n$. Further, we can then view the $\mathbf{B}$ maintained by the Frequent Directions algorithm as an array of $m$ buckets.

Now the setting is exactly the same as the Misra-Gries algorithm. Consequently, the constructive lower bound proof from Theorem 3.1 directly applies to Frequent Directions.

$\square$

## D.2 Learned Frequent Directions Under Zipfian

We need an additional result to prove Theorem 3.4. Recall that $\mathbf{P}_H$ in Algorithm 2 consists of orthonormal column vectors $\vec{w}_1, \dots, \vec{w}_{m_L} \in \mathbb{R}^d$. Extending this set of vectors to form an orthonormal basis of $\mathbb{R}^d$: $\vec{w}_1, \dots, \vec{w}_{m_L}, \vec{w}_{m_L+1}, \dots, \vec{w}_d$. Write $\mathbf{P}_{\overline{H}} = [\vec{w}_{m_L+1} \mid \dots \mid \vec{w}_d]$ the projection matrix to the orthogonal subspace. Let $\mathbf{A}_\downarrow \doteq \mathbf{A}\mathbf{P}_H\mathbf{P}_H^T$ be the matrix of projecting the rows of $\mathbf{A}$ to the predicted subspace, and $\mathbf{A}_\perp \doteq \mathbf{A} - \mathbf{A}_\downarrow = \mathbf{A}(\boldsymbol{I} - \mathbf{P}_H\mathbf{P}_H^T)$.

The following lemma holds.

**Lemma D.1.** *For a vector $\vec{x} \in \mathbb{R}^d$, we have*

$$\vec{x}^T \mathbf{A}^T \mathbf{A} \vec{x} = \vec{x}^T \mathbf{A}_\downarrow^T \mathbf{A}_\downarrow \vec{x} + \vec{x}^T \mathbf{A}_\perp^T \mathbf{A}_\perp \vec{x} + 2 \cdot \sum_{i \in [d]} \sigma_i^2 \cdot \langle \mathbf{P}_H^T \vec{v}_i, \mathbf{P}_H^T \vec{x} \rangle \cdot \langle \mathbf{P}_{\overline{H}}^T \vec{v}_i, \mathbf{P}_{\overline{H}}^T \vec{x} \rangle. \quad (46)$$

The proof of the lemma is included at the end of the section.

**Proof of Theorem 3.4.** We need to prove both upper bounds and lower bounds for the theorem.

*Upper Bound for Theorem 3.4.* It suffices to show that, there exists a parameter setting of $m_L$ which enables the algorithm to achieve the desired error bound. We assume that the algorithm uses $m_L = m/3$ predicted directions from the learned oracle.

Recall that Algorithm 2 maintains two instances of Algorithm 1: $\mathcal{A}_{FD}^\downarrow$ and $\mathcal{A}_{FD}^\perp$. The former processes the vectors projected onto the subspace defined by $\mathbf{P}_H$, while the latter handles the vectors projected onto the orthogonal subspace. Therefore, the input to $\mathcal{A}_{FD}^\downarrow$ is $\mathbf{A}_\downarrow = \mathbf{A}\mathbf{P}_H\mathbf{P}_H^T$, and the input to $\mathcal{A}_{FD}^\perp$ is $\mathbf{A}_\perp = \mathbf{A} - \mathbf{A}_\downarrow = \mathbf{A}(\boldsymbol{I} - \mathbf{P}_H\mathbf{P}_H^T) = \mathbf{A}\mathbf{P}_{\overline{H}}\mathbf{P}_{\overline{H}}^T$. Ultimately, the resulting matrix $\mathbf{B}$ is a combination of the matrices returned by $\mathcal{A}_{FD}^\downarrow$ and $\mathcal{A}_{FD}^\perp$, specifically denoted as $\mathbf{B}_\downarrow$ and $\mathbf{B}_\perp$, respectively.

Combined with Lemma D.1, for each right singular vector $\vec{v}_j$ of $\mathbf{A}$, we have

$$\vec{v}_j^T (\mathbf{A}^T \mathbf{A} - \mathbf{B}^T \mathbf{B}) \vec{v}_j = \vec{v}_j^T \mathbf{A}^T \vec{v}_j \mathbf{A} - \vec{v}_j^T (\mathbf{B}_\downarrow)^T \mathbf{B}_\downarrow \vec{v}_j - \vec{v}_j^T (\mathbf{B}_\perp)^T \mathbf{B}_\perp \vec{v}_j \quad (47)$$

$$= \vec{v}_j^T \mathbf{A}_\downarrow^T \mathbf{A}_\downarrow \vec{v}_j - \vec{v}_j^T (\mathbf{B}_\downarrow)^T \mathbf{B}_\downarrow \vec{v}_j \quad (48)$$

$$+ \vec{v}_j^T \mathbf{A}_\perp^T \mathbf{A}_\perp \vec{v}_j - \vec{v}_j^T (\mathbf{B}_\perp)^T \mathbf{B}_\perp \vec{v}_j \quad (49)$$

$$+ 2 \cdot \sum_{i \in [d]} \sigma_i^2 \cdot \langle \mathbf{P}_H^T \vec{v}_i, \mathbf{P}_H^T \vec{v}_j \rangle \cdot \langle \mathbf{P}_{\overline{H}}^T \vec{v}_i, \mathbf{P}_{\overline{H}}^T \vec{v}_j \rangle. \quad (50)$$

First, observe that since $\mathcal{A}_{FD}^\downarrow$ is allocated $m_L \times d$ space for the matrix $\mathbf{A}^\downarrow$ with rank at most $m_L$, by the error guarantee of Frequent Direction algorithm (Fact 2.2), it is guaranteed that $\vec{v}_j^T \mathbf{A}_\downarrow^T \mathbf{A}_\downarrow \vec{v}_j - \vec{v}_j^T (\mathbf{B}_\downarrow)^T \mathbf{B}_\downarrow \vec{v}_j = 0$.

Second, note that $\langle \mathbf{P}_H^T \vec{v}_i, \mathbf{P}_H^T \vec{v}_j \rangle$ is the inner product, between the projected vectors $\vec{v}_i$ and $\vec{v}_j$ to the subspace $H$ specified by the predicted frequent directions, and that $\langle \mathbf{P}_{\overline{H}}^T \vec{v}_i, \mathbf{P}_{\overline{H}}^T \vec{v}_j \rangle$ is the inner product, between the projected vectors $\vec{v}_i$ and $\vec{v}_j$ to the orthogonal complement of $H$.

In particular, when the machine learning oracle makes perfect predictions of $\vec{v}_1, \ldots, \vec{v}_{m_L}$, i.e., $\vec{w}_1 = \vec{v}_1, \ldots, \vec{w}_{m_L} = \vec{v}_{m_L}$, then for each $i$, either $\mathbf{P}_H^T \vec{v}_i$ or $\mathbf{P}_{\bar{H}}^T \vec{v}_i$ will be zero.

Therefore, it holds that

$$\vec{v}_j^T (\mathbf{A}^T \mathbf{A} - \mathbf{B}^T \mathbf{B}) \vec{v}_j = \vec{v}_j^T \mathbf{A}_\perp^T \mathbf{A}_\perp \vec{v}_j - \vec{v}_j^T (\mathbf{B}_\perp)^T \mathbf{B}_\perp \vec{v}_j. \tag{51}$$

Further, by the property of Frequent Direction algorithm $\mathcal{A}_{FD}^\perp$, $\mathbf{A}_\perp^T \mathbf{A}_\perp - (\mathbf{B}_\perp)^T \mathbf{B}_\perp \succeq 0$. And since $\mathbf{A}_\perp$ is the projection of $\mathbf{A}$ to the subspace spanned by the right singular vectors $\vec{v}_{m_L+1}, \ldots, \vec{v}_d$, it still has right singular vectors $\vec{v}_{m_L+1}, \ldots, \vec{v}_d$, associated with singular values $\sigma_{m_L+1}, \ldots, \sigma_d$. It follows that

$$0 \le \vec{v}_j^T (\mathbf{A}^T \mathbf{A} - \mathbf{B}^T \mathbf{B}) \vec{v}_j \tag{52}$$

$$= \vec{v}_j^T \mathbf{A}_\perp^T \mathbf{A}_\perp \vec{v}_j - \vec{v}_j^T (\mathbf{B}_\perp)^T \mathbf{B}_\perp \vec{v}_j \tag{53}$$

$$\le \vec{v}_j^T \mathbf{A}_\perp^T \mathbf{A}_\perp \vec{v}_j \tag{54}$$

$$\le \begin{cases} \sigma_j^2, & j > m_L \\ 0 & j \le m_L \end{cases}. \tag{55}$$

Therefore, the weighted error is given by

$$\mathcal{E}rr(\mathcal{A}_{FD}) \doteq \sum_{i \in [d]} \frac{\sigma_i^2}{\|\mathbf{A}\|_F^2} \cdot \vec{v}_i^T (\mathbf{A}^T \mathbf{A} - \mathbf{B}^T \mathbf{B}) \vec{v}_i \tag{56}$$

$$= \sum_{i=1}^{m_L} \frac{\sigma_i^2}{\|\mathbf{A}\|_F^2} \cdot \vec{v}_i^T (\mathbf{A}^T \mathbf{A} - \mathbf{B}^T \mathbf{B}) \vec{v}_i + \sum_{i=m_L+1}^{d} \frac{\sigma_i^2}{\|\mathbf{A}\|_F^2} \cdot \vec{v}_i^T (\mathbf{A}^T \mathbf{A} - \mathbf{B}^T \mathbf{B}) \vec{v}_i \tag{57}$$

$$= \sum_{i=m_L+1}^{d} \frac{\sigma_i^2}{\|\mathbf{A}\|_F^2} \cdot \vec{v}_i^T (\mathbf{A}^T \mathbf{A} - \mathbf{B}^T \mathbf{B}) \vec{v}_i \tag{58}$$

$$\in O\left( \sum_{i=m_L+1}^{d} \frac{\sigma_i^2}{\|\mathbf{A}\|_F^2} \cdot \sigma_i^2 \right) \tag{59}$$

$$= O\left( \sum_{i=m_L+1}^{d} \frac{1}{i \cdot \ln d} \cdot \frac{\|\mathbf{A}\|_F^2}{i \cdot \ln d} \right) \tag{60}$$

$$= O\left( \frac{1}{m_L + 1} \cdot \frac{\|\mathbf{A}\|_F^2}{(\ln d)^2} \right) \tag{61}$$

Noting that $m_L \in \Theta(m)$ finishes the proof of upper bound.

*Lower Bound for Theorem 3.4.* The proof of the lower bound follows the same approach as the one for Theorem 3.2, in Appendix C.

Assume that $\mathbf{A}$ consists of standard basis vectors $\vec{e}_1, \ldots, \vec{e}_d \in \mathbb{R}^d$. Let $f(\vec{e}_i)$ denote the number of occurrences of $\vec{e}_i$ in $\mathbf{A}$. Without loss of generality, assume that $f(\vec{e}_1) \ge \ldots \ge f(\vec{e}_d)$. In this case, we have $f(\vec{e}_i) = \sigma_i^2$ and $\sum_{i \in [d]} f(\vec{e}_i) = \sum_{i \in [d]} \sigma_i^2 = \|\mathbf{A}\|_F^2 = n$. Further, we can then view the $\mathbf{B}$ maintained by the Frequent Directions algorithm as an array of $m$ buckets.

Now the setting is exactly the same as the Misra-Gries algorithm. Consequently, the constructive lower bound proof from Theorem 3.2 directly applies to learned Frequent Directions.

$\square$

We next prove Lemma D.1.

*Proof of Lemma D.1.* First, observe that

$$\vec{x}^T \mathbf{A}^T \mathbf{A} \vec{x} = \vec{x}^T (\mathbf{A}_\downarrow + \mathbf{A}_\perp)^T (\mathbf{A}_\downarrow + \mathbf{A}_\perp) \vec{x} \tag{62}$$

$$= \vec{x}^T \mathbf{A}_\downarrow^T \mathbf{A}_\downarrow \vec{x} + \vec{x}^T \mathbf{A}_\perp^T \mathbf{A}_\perp \vec{x} + \vec{x}^T \mathbf{A}_\downarrow^T \mathbf{A}_\perp \vec{x} + \vec{x}^T \mathbf{A}_\perp^T \mathbf{A}_\downarrow \vec{x}. \tag{63}$$

It suffices to show that

$$\vec{x}^T \mathbf{A}_\downarrow^T \mathbf{A}_\perp \vec{x} = \vec{x}^T \mathbf{A}_\perp^T \mathbf{A}_\downarrow \vec{x} = \sum_{i \in [d]} \sigma_i^2 \cdot \langle \mathbf{P}_{\overline{H}}^T \vec{v}_i, \mathbf{P}_{\overline{H}}^T \vec{x} \rangle \cdot \langle \mathbf{P}_H^T \vec{v}_i, \mathbf{P}_H^T \vec{x} \rangle \tag{64}$$

Note that

$$\vec{x}^T \mathbf{A}_\downarrow^T \mathbf{A}_\perp \vec{x} = \vec{x}^T \left( \mathbf{P}_H^T \mathbf{P}_H \mathbf{A}^T \right) \left( \mathbf{A}(\boldsymbol{I} - \mathbf{P}_H \mathbf{P}_H^T) \vec{x} \right) \tag{65}$$

$$= \vec{x}^T \mathbf{P}_H^T \mathbf{P}_H \mathbf{A}^T \mathbf{A}(\boldsymbol{I} - \mathbf{P}_H \mathbf{P}_H^T) \vec{x} \tag{66}$$

$$= \vec{x}^T (\boldsymbol{I} - \mathbf{P}_H \mathbf{P}_H^T)^T \mathbf{A}^T \mathbf{A} \mathbf{P}_H \mathbf{P}_H^T \vec{x} = \vec{x}^T \mathbf{A}_\perp^T \mathbf{A}_\downarrow \vec{x}. \tag{67}$$

Hence, it remains to study $\vec{x}^T \mathbf{A}_\downarrow^T \mathbf{A}_\perp \vec{x}$ or $\vec{x}^T \mathbf{A}_\perp^T \mathbf{A}_\downarrow \vec{x}$. Keeping expanding one of them

$$\vec{x}^T \mathbf{A}_\downarrow^T \mathbf{A}_\perp \vec{x} = \vec{x}^T (\boldsymbol{I} - \mathbf{P}_H \mathbf{P}_H^T)^T \mathbf{A}^T \mathbf{A} \mathbf{P}_H \mathbf{P}_H^T \vec{x} \tag{68}$$

$$= \vec{x}^T (\boldsymbol{I} - \mathbf{P}_H \mathbf{P}_H^T)^T \left( \sum_{i \in [d]} \sigma_i^2 \vec{v}_i \vec{v}_i^T \right) \mathbf{P}_H \mathbf{P}_H^T \vec{x} \tag{69}$$

$$= \sum_{i \in [d]} \sigma_i^2 \cdot \langle \vec{v}_i, (\boldsymbol{I} - \mathbf{P}_H \mathbf{P}_H^T) \vec{x} \rangle \cdot \langle \vec{v}_i, \mathbf{P}_H \mathbf{P}_H^T \vec{x} \rangle. \tag{70}$$

Since $\boldsymbol{I} - \mathbf{P}_H \mathbf{P}_H^T = \mathbf{P}_{\overline{H}} \mathbf{P}_{\overline{H}}^T$,

$$\vec{x}^T \mathbf{A}_\downarrow^T \mathbf{A}_\perp \vec{x} = \sum_{i \in [d]} \sigma_i^2 \cdot \langle \vec{v}_i, \mathbf{P}_{\overline{H}} \mathbf{P}_{\overline{H}}^T \vec{x} \rangle \cdot \langle \vec{v}_i, \mathbf{P}_H \mathbf{P}_H^T \vec{x} \rangle \tag{71}$$

$$= \sum_{i \in [d]} \sigma_i^2 \cdot \langle \mathbf{P}_{\overline{H}}^T \vec{v}_i, \mathbf{P}_{\overline{H}}^T \vec{x} \rangle \cdot \langle \mathbf{P}_H^T \vec{v}_i, \mathbf{P}_H^T \vec{x} \rangle \tag{72}$$

$$\square$$

## E  CONSISTENCY/ROBUSTNESS TRADE OFFS

If the predictions are perfect, the sketch $\mathbf{B}$ output by Algorithm 2 satisfies that $\mathbf{B}^T\mathbf{B} \preceq \mathbf{A}^T\mathbf{A}$. This property in particular gives the quantity $\vec{x}^T\mathbf{B}^T\mathbf{B}\vec{x} \leq \vec{x}^T\mathbf{A}^T\mathbf{A}\vec{x}$ for any vector $\vec{x}$ and as further $\mathbf{B}^T\mathbf{B} \succeq 0$, we get that the error $|\vec{v}_i^T\mathbf{B}^T\mathbf{B}\vec{v}_i - \vec{v}_i^T\mathbf{A}^T\mathbf{A}\vec{v}_i| \leq \sigma_i^2$ for any of the singular vectors $\vec{v}_i$ (and with perfect predictions the errors on the predicted singular vectors are zero). Unfortunately, with imperfect predictions, the guarantee that $\mathbf{B}^T\mathbf{B} \preceq \mathbf{A}^T\mathbf{A}$ is not retained. To take a simple example, suppose that $d = 2$ and that the input matrix $\mathbf{A} = (1,1)$ has just one row. Suppose we create two frequent direction sketches by projecting onto the standard basis vectors $e_1$ and $e_2$ and stack the resulting sketches $\mathbf{B}_1$ and $\mathbf{B}_2$ to get a sketch matrix $\mathbf{B}$. It is then easy to check that $\mathbf{B}$ is in fact the identify matrix. In particular, if $\vec{x} = e_1 - e_2$, then $\|\mathbf{B}\vec{x}\|_2^2 = 2$ whereas $\|\mathbf{A}\vec{x}\|_2^2 = 0$ showing that $\mathbf{A}^T\mathbf{A} - \mathbf{B}^T\mathbf{B}$ is not positive semidefinite. The absence of this property poses issues in proving consistency/robustness trade offs for the algorithm. Indeed, our analysis of the classic frequent directions algorithm under Zipfian distributions, crucially uses that the error incurred in the light directions $\vec{v}_i$ for $i \geq \frac{m}{\ln\frac{d}{m}}$ is at most $\sigma_i^2$.

In this section, we address this issue by presenting a variant of Algorithm 2 that does indeed provide consistency/robustness trade-offs with only a constant factor blow up in space. To do so, we will maintain three different sketches of the matrix $\mathbf{A}$. The first sketch is the standard frequent directions sketch Liberty (2013) in Algorithm 1, the second one is the learning-augmented sketch produced by Algorithm 2, and the final sketch computes an approximation to the residual error $\|\mathbf{A} - [\mathbf{A}]_k\|_F^2$ within a constant factor using an algorithm from Li et al. (2024). Let $\mathbf{B}_1$ be the output of Algorithm 1 on input $\mathbf{A}$ and $\mathbf{B}_2$ be the output of Algorithm 2 on input $\mathbf{A}$. Suppose for simplicity that we knew $\|\mathbf{A} - [\mathbf{A}]_k\|_F^2$ exactly. Then, the idea is that when queried with a unit vector $\vec{x}$, we compute $\|\mathbf{B}_1 x\|_2^2$ and $\|\mathbf{B}_2 x\|_2^2$. If these are within $2\frac{\|\mathbf{A}-[\mathbf{A}]_k\|_F^2}{m-k}$ of each other, we output $\|\mathbf{B}_2 x\|_2^2$ as the final estimate of $\vec{x}^T\mathbf{A}^T\mathbf{A}\vec{x}$, otherwise, we output $\|\mathbf{B}_1 x\|_2^2$. The idea behind this approach is that in the latter case, we know that the learning-based algorithm must have performed poorly with an error of at least $\frac{\|\mathbf{A}-[\mathbf{A}]_k\|_F^2}{m-k}$ and by outputting the estimate from the classic algorithm, we retain its theoretical guarantee. On the other hand, in the former case, we know that the error is at most $3\frac{\|\mathbf{A}-[\mathbf{A}]_k\|_F^2}{m-k}$ but could be much better if the learning augmented algorithm performed well.

To state our the exact result, we recall that the algorithm from Li et al. (2024) using space $O(k^2/\varepsilon^4)$ maintains a sketch of $\mathbf{A}$ such that from the sketch we can compute an estimate $\alpha$ such that $\|\mathbf{A} - [\mathbf{A}]_k\|_F^2 \leq \alpha \leq (1+\varepsilon)\|\mathbf{A} - [\mathbf{A}]_k\|_F^2$. We denote this algorithm $\mathcal{A}_{res}(k, \varepsilon)$. Our final algorithm is Algorithm 3 for which we prove the following result.

**Theorem E.1.** *[Worst-Case guarantees] For any unit vector $\vec{x}$, the estimate $\Gamma$ of $\|\mathbf{A}\vec{x}\|_2^2$ returned by Algorithm 3 satisfies*

$$\left|\|\mathbf{A}\vec{x}\|_2^2 - \Gamma\right| \leq \min\left(\left|\|\mathbf{A}\vec{x}\|_2^2 - \|\mathbf{B}_2 x\|_2^2\right|, 6\frac{\|\mathbf{A} - [\mathbf{A}]_k\|_F^2}{m-k}\right).$$

*In other words, the Error of Algorithm 3 is asymptotically bounded by the minimum of Algorithm 2 and the classic Frequent Direction algorithm.*

*Proof.* Suppose first that $\left|\|\mathbf{B}_2\vec{x}\|_2^2 - \|\mathbf{B}_1\vec{x}\|_2^2\right| \leq 2\alpha$. Then $\left|\|\mathbf{A}\vec{x}\|_2^2 - \Gamma\right| = \left|\|\mathbf{A}\vec{x}\|_2^2 - \|\mathbf{B}_2 x\|_2^2\right|$. Moreover, by the approximation guarantees of $\mathcal{A}_{res}$ and $\mathcal{A}_{FD}$,

$$\left|\|\mathbf{A}\vec{x}\|_2^2 - \|\mathbf{B}_2 x\|_2^2\right| \leq \left|\|\mathbf{A}\vec{x}\|_2^2 - \|\mathbf{B}_1 x\|_2^2\right| + \left|\|\mathbf{B}_1 x\|_2^2 - \|\mathbf{B}_2 x\|_2^2\right| \leq \alpha + 2\alpha \leq 6\frac{\|\mathbf{A} - [\mathbf{A}]_k\|_F^2}{m-k},$$

as desired.

Suppose on the other hand that $\left|\|\mathbf{B}_2\vec{x}\|_2^2 - \|\mathbf{B}_1\vec{x}\|_2^2\right| > 2\alpha$. Since by Fact 2.2, we always have that $\left|\|\mathbf{A}\vec{x}\|_2^2 - \|\mathbf{B}_1 x\|_2^2\right| \leq \frac{\|\mathbf{A}-[\mathbf{A}]_k\|_F^2}{m-k} \leq \alpha$, it follows that $\left|\|\mathbf{A}\vec{x}\|_2^2 - \|\mathbf{B}_2 x\|_2^2\right| > \alpha \geq \frac{\|\mathbf{A}-[\mathbf{A}]_k\|_F^2}{m-k}$. But since in this case, we output $\|\mathbf{B}_1\vec{x}\|_2^2$, the estimate of the standard frequent direction, we again have by Fact 2.2 that $\left|\|\mathbf{A}\vec{x}\|_2^2 - \|\mathbf{B}_2 x\|_2^2\right| \leq \frac{\|\mathbf{A}-[\mathbf{A}]_k\|_F^2}{m-k}$ as desired. □

We note that the constant 6 in the theorem can be replaced by any constant $> 3$ by increasing the space used for $\mathcal{A}_{res}$.

---

**Algorithm 3** Robust Learning-based Frequent Direction $\mathcal{A}_{RLFD}$

---

1: **Procedure** INITIALIZATION
2:    **Input:** sketch parameters $m, d \in \mathbb{N}^+$; learned oracle parameter $m_L$ s.t., $m_L \leq m$; predicted frequent directions $\mathbf{P}_H = [\vec{w}_1 \mid \ldots \mid \vec{w}_{m_L}] \in \mathbb{R}^{d \times m_L}$, query vector $\vec{x}$.
3:    Initialize an instance of Algorithm 1: $\mathcal{A}_{FD}.initialization(m, 0.5 \cdot m, d)$
4:    Initialize an instance of Algorithm 2: $\mathcal{A}_{LFD}.initialization(m, 0.5 \cdot m, d)$
5:    Initialize the residual error estimation algorithm Li et al. (2024) $\mathcal{A}_{res}(m/2, 1)$

6: **Procedure** UPDATE
7:    $\mathcal{A}_{FD}.update(\mathbf{A}_i)$
8:    $\mathcal{A}_{LFD}.update(\mathbf{A}_i)$
9:    $\mathcal{A}_{res}.update(\mathbf{A}_i)$

10: **Procedure** RETURN
11:    $\mathbf{B}_1 \leftarrow \mathcal{A}_{FD}.return()$
12:    $\mathbf{B}_2 \leftarrow \mathcal{A}_{FD}.return()$
13:    $\alpha_0 \leftarrow \mathcal{A}_{res}.return()$
14:    $\alpha \leftarrow \frac{\alpha_0}{m-k}$
15:    **return** $(\mathbf{B}_1, \mathbf{B}_2, \alpha)$

16: **Procedure** QUERY($\vec{x}$)
17:    **if** $\big| \|\mathbf{B}_2\vec{x}\|_2^2 - \|\mathbf{B}_1\vec{x}\|_2^2 \big| \leq 2\alpha$ **then**
18:       **return** $\|\mathbf{B}_2\vec{x}\|_2^2$
19:    **else**
20:       **return** $\|\mathbf{B}_1\vec{x}\|_2^2$

---

## E.1   THE ERROR OF NON-PERFECT ORACLES.

We will now obtain a more fine-grained understanding of the consistency/robustness trade off of Algorithm 3. Consider the SVD $\mathbf{A} = \sum_{i \in [d]} \sigma_i \vec{u}_i \vec{v}_i^T$. Let $\mathbf{A}_\downarrow \doteq \mathbf{A} \mathbf{P}_H \mathbf{P}_H^T$ be the matrix of projecting the rows of $\mathbf{A}$ to the predicted subspace, and $\mathbf{A}_\perp \doteq \mathbf{A} - \mathbf{A}_\downarrow = \mathbf{A}(\mathbf{I} - \mathbf{P}_H \mathbf{P}_H^T)$. Recall that $\mathbf{P}_H$ consists of orthonormal column vectors $\vec{w}_1, \ldots, \vec{w}_{m_L} \in \mathbb{R}^d$. Extending this set of vectors to form an orthonormal basis of $\mathbb{R}^d$: $\vec{w}_1, \ldots, \vec{w}_{m_L}, \vec{w}_{m_L+1}, \ldots, \vec{w}_d$. Write $\mathbf{P}_{\overline{H}} = [\vec{w}_{m_L+1} \mid \ldots \mid \vec{w}_d]$ the projection matrix to the orthogonal subspace.

Based on Lemma D.1, for each vector, we can write

$$\vec{x}^T \mathbf{A}^T \mathbf{A} \vec{x} = \vec{x}^T \mathbf{A}_\downarrow^T \mathbf{A}_\downarrow \vec{x} + \vec{x}^T \mathbf{A}_\perp^T \mathbf{A}_\perp \vec{x} + 2 \cdot \sum_{i \in [d]} \sigma_i^2 \cdot \langle \mathbf{P}_H^T \vec{v}_i, \mathbf{P}_H^T \vec{x} \rangle \cdot \langle \mathbf{P}_{\overline{H}}^T \vec{v}_i, \mathbf{P}_{\overline{H}}^T \vec{x} \rangle. \tag{73}$$

To understand the significance of Lemma D.1, note that our algorithm attempts to approximate the first two terms (through either exact or approximate Frequent Direction sketches), but ignores the final one. Therefore, regardless of how successful it is in approximating $\vec{x}^T \mathbf{A}_\downarrow^T \mathbf{A}_\downarrow \vec{x} + \vec{x}^T \mathbf{A}_\perp^T \mathbf{A}_\perp \vec{x}$, we will have $2 \cdot \sum_{i \in [d]} \sigma_i^2 \cdot \langle \mathbf{P}_H^T \vec{v}_i, \mathbf{P}_H^T \vec{x} \rangle \cdot \langle \mathbf{P}_{\overline{H}}^T \vec{v}_i, \mathbf{P}_{\overline{H}}^T \vec{x} \rangle$ occurring as an additional added error.

Note that $\langle \mathbf{P}_H^T \vec{v}_i, \mathbf{P}_H^T \vec{v}_j \rangle$ is the inner product, between the projected vectors $\vec{v}_i$ and $\vec{v}_j$ to the subspace $H$ specified by the predicted frequent directions, and that $\langle \mathbf{P}_{\overline{H}}^T \vec{v}_i, \mathbf{P}_{\overline{H}}^T \vec{v}_j \rangle$ is the inner product, between the projected vectors $\vec{v}_i$ and $\vec{v}_j$ to the orthogonal complement of $H$. In particular, if $\mathbf{P}_H$ consists of a set of correctly predicted singular vectors of $\mathbf{A}$, then for any $i$, either $\mathbf{P}_H^T \vec{v}_i$ or $\mathbf{P}_{\overline{H}}^T \vec{v}_i$ will be zero and in particular the additional added error will be zero. In order to obtain an algorithm performing as well as if we had perfect predictions, it therefore suffices that the predictions are accurate enough that

$$\left| \sum_{i \in [d]} \sigma_i^2 \cdot \langle \mathbf{P}_H^T \vec{v}_i, \mathbf{P}_H^T \vec{v}_j \rangle \cdot \langle \mathbf{P}_{\overline{H}}^T \vec{v}_i, \mathbf{P}_{\overline{H}}^T \vec{v}_j \rangle \right| \in O\left( \frac{\|\mathbf{A} - [\mathbf{A}]_k\|_F^2}{m}. \right) \tag{74}$$

To obtain a more general smoothness/robustness trade off, one can plug into Theorem E.1. Doing so in the setting of Theorem 3.4 where the singular values follow a Zipfian distribution, we obtain the following immediate corollary.

**Corollary E.2.** *Consider the setting of Theorem 3.4, but where we run Algorithm 3 instead of Algorithm 2 and where we make no assumptions on the quality of the oracle. Then the error $\mathcal{E}rr(\mathcal{A}_{RLFD})$ is at most*

$$O\left(\frac{1}{(\ln d)^2} \cdot \frac{\|\mathbf{A}\|_F^2}{m}\right) + 2\sum_{i\in[d]} \frac{\sigma_i^2}{\|\mathbf{A}\|_F^2} \cdot \sum_{j\in[d]} \sigma_j^2 \cdot \langle \mathbf{P}_H^T \vec{v}_j, \mathbf{P}_H^T \vec{v}_i \rangle \cdot \langle \mathbf{P}_{\overline{H}}^T \vec{v}_j, \mathbf{P}_{\overline{H}}^T \vec{v}_i \rangle,$$

*but also always bounded by*

$$O\left(\frac{\left(\ln \frac{m}{\ln \frac{d}{m}}\right) \cdot \ln \frac{d}{m}}{(\ln d)^2} \cdot \frac{\|\mathbf{A}\|_F^2}{m}\right).$$

We finish by showing an example demonstrating that even with very accurate predictions, the extra added error can be prohibitive. Assume that the input space is $\mathbb{R}^2$, and the input vectors are either $(1,0)$ or $(0,1)$. Assume that $\sigma_1^2 = 10^7$, $\sigma_2^2 = 1$,, $\vec{v}_1 = (1,0)$, and $\vec{v}_2 = (0,1)$.

In this case, assume that $m_L = 1$. A perfect $\mathbf{P}_H$ should be $\mathbf{P}_H = (1,0)$, but we will assume that the actual prediction we get is a little perturbed, say we change it to $\mathbf{P}_H = (\cos\frac{1}{100}, \sin\frac{1}{100})$. Therefore, $\mathbf{P}_{\overline{H}} = (\sin\frac{1}{100}, -\cos\frac{1}{100})$,

$$\sum_{i\in[2]} \sigma_i^2 \cdot \langle \mathbf{P}_H^T \vec{v}_i, \mathbf{P}_H^T \vec{v}_1 \rangle \cdot \langle \mathbf{P}_{\overline{H}}^T \vec{v}_i, \mathbf{P}_{\overline{H}}^T \vec{v}_1 \rangle = 10^7 \cdot \langle \cos\frac{1}{100}, \cos\frac{1}{100} \rangle \cdot \langle \sin\frac{1}{100}, \sin\frac{1}{100} \rangle \quad (75)$$

$$+ 1 \cdot \langle \sin\frac{1}{100}, \cos\frac{1}{100} \rangle \cdot \langle -\cos\frac{1}{100}, \sin\frac{1}{100} \rangle \quad (76)$$

$$\approx 10^7 \cos^2\frac{1}{100} \cdot \sin^2\frac{1}{100} \quad (77)$$

$$\approx 10^7 \cdot \frac{1}{100^2} \approx 10^3. \quad (78)$$

In general, assume that $\mathbf{P}_H = (\cos\theta, \sin\theta)$ for small $\theta$. The

$$\sum_{i\in[2]} \sigma_i^2 \cdot \langle \mathbf{P}_H^T \vec{v}_i, \mathbf{P}_H^T \vec{v}_1 \rangle \cdot \langle \mathbf{P}_{\overline{H}}^T \vec{v}_i, \mathbf{P}_{\overline{H}}^T \vec{v}_1 \rangle = \sigma_1^2 \cos^2\theta \cdot \sin^2\theta \approx \sigma_1^2 \theta^2 \quad (79)$$

So we need $\theta \approx 1/\sqrt{m}$, in order that this bound is comparable with the normal FD bound.

## F    ADDITIONAL EXPERIMENTS

In this section, we include figures which did not fit in the main text.

### F.1    DATASET STATISTICS

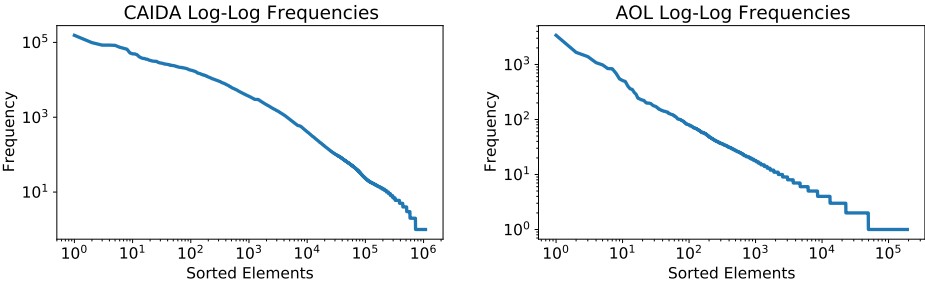

Figure 3: Log-log plot of frequencies for the CAIDA and AOL datasets.

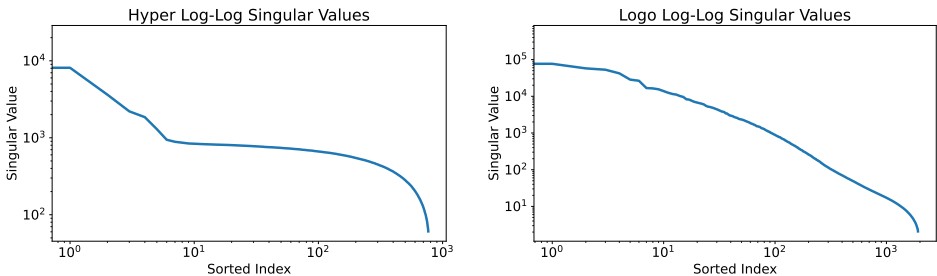

Figure 4: Log-log plot of singular values for the first Hyper and Logo matrices.

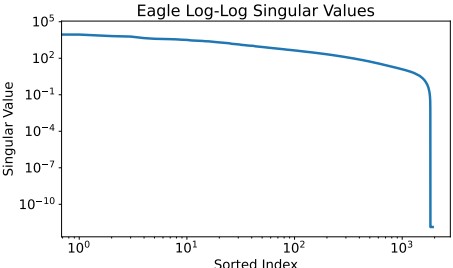

Figure 5: Log-log plot of singular values for the first Eagle and Friends matrices.

### F.2    NOISE ANALYSIS IN FREQUENT DIRECTIONS

We present the following figure for the

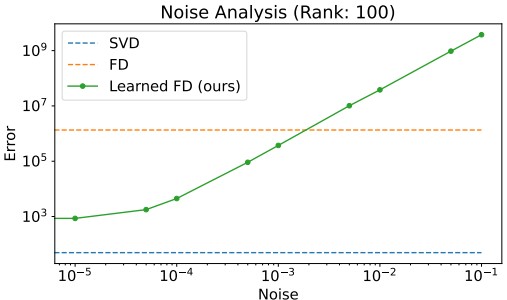

Figure 6: Analysis of prediction noise in matrix streaming on the first matrix of the Logo dataset. The rank of the algorithms is 100. The baselines of Frequent Directions and the true SVD are shown as dashed lines. Our learned Frequent Directions algorithm uses perfect predictions corrupted by a matrix of Gaussian noise with standard deviation $\sigma/\sqrt{d}$ where $\sigma$ is displayed as the amount of noise on the horizontal axis. The linear relationship on the log-log plot indicates that the performance of our algorithm decays polynomially with the amount of noise.

### F.3 ADDITIONAL FREQUENT DIRECTIONS EXPERIMENTS

We present plots of error/rank tradeoffs and error across sequences of matrices with fixed rank for all four datasets Hyper, Logo, Eagle, and Friends.

#### F.3.1 HYPER DATASET

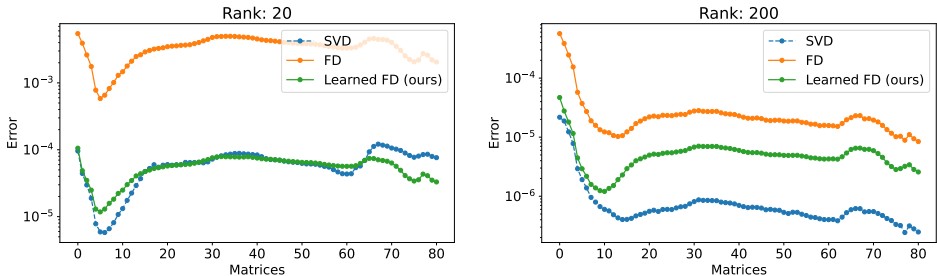

Figure 7: Frequent directions results on the Hyper dataset.

#### F.3.2 LOGO DATASET

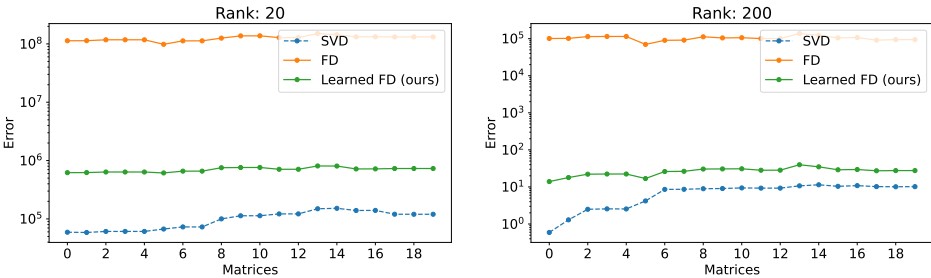

Figure 8: Frequent directions results on the Logo dataset.

### F.3.3 EAGLE DATASET

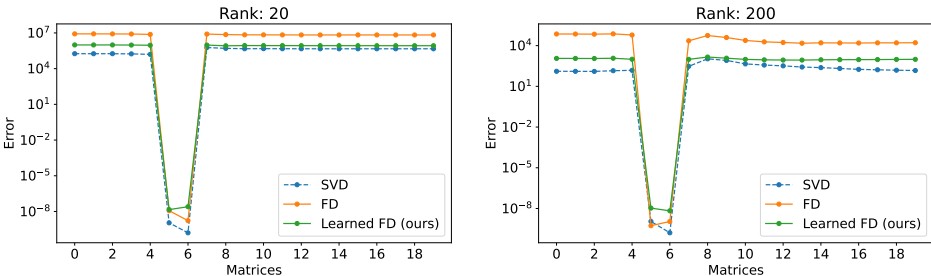

Figure 9: Frequent directions results on the Eagle dataset.

### F.3.4 FRIENDS DATASET

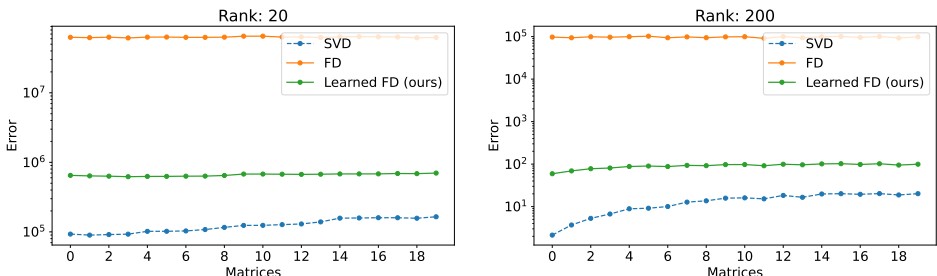

Figure 10: Frequent directions results on the Friends dataset.

## F.4 ADDITIONAL FREQUENCY ESTIMATION EXPERIMENTS

Here, we present all frequency estimation results comparing our Learned Misra-Gries algorithm with Learned CountSketch of Hsu et al. (2019) and Learned CountSketch++ of Aamand et al. (2023). We present results both with and without learned predictions. Additionally, we present results both with standard weighted error discussed in this paper as well as unweighted error also evaluated in the experiments of prior work. The unweighted error corresponds to taking the sum of absolute errors across all items appearing in the stream (not weighted by their frequencies).

### F.4.1 NO PREDICTIONS, WEIGHTED ERROR

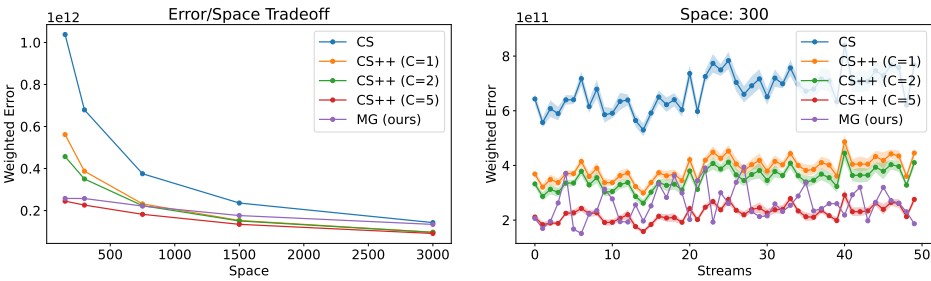

Figure 11: Frequency estimation on the CAIDA dataset with weighted error and no predictions.

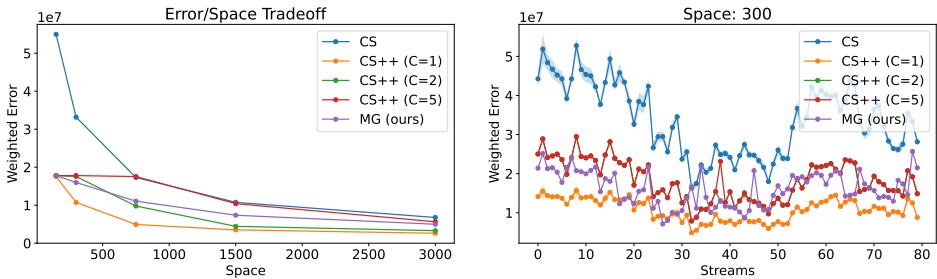

Figure 12: Frequency estimation on the AOL dataset with weighted error and no predictions.

### F.4.2   WITH PREDICTIONS, WEIGHTED ERROR

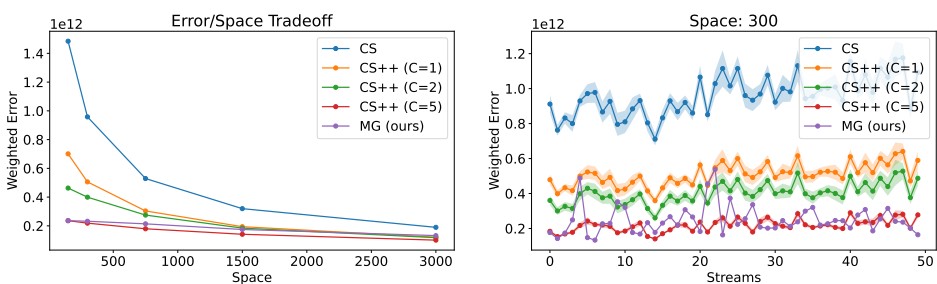

Figure 13: Frequency estimation on the CAIDA dataset with weighted error and learned predictions.

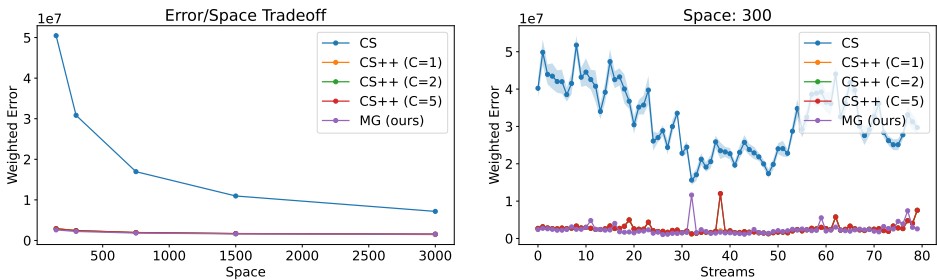

Figure 14: Frequency estimation on the AOL dataset with weighted error and learned predictions.

### F.4.3   NO PREDICTIONS, UNWEIGHTED ERROR

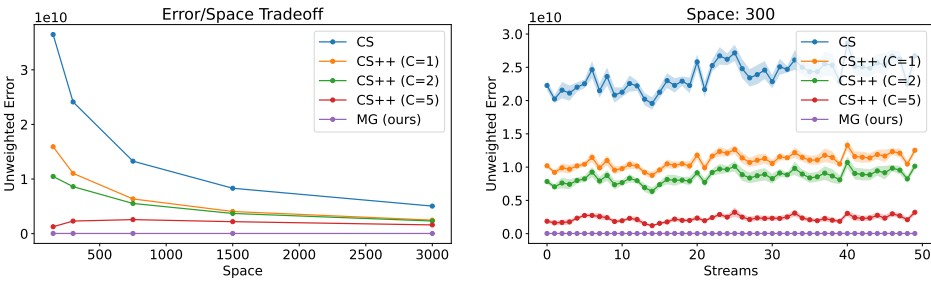

Figure 15: Frequency estimation on the CAIDA dataset with unweighted error and no predictions.

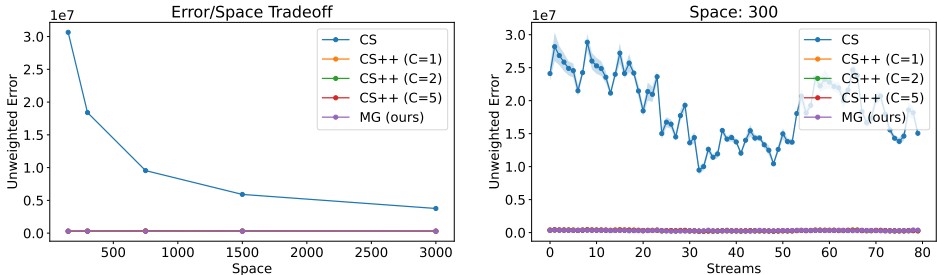

Figure 16: Frequency estimation on the AOL dataset with unweighted error and no predictions.

### F.4.4 WITH PREDICTIONS, UNWEIGHTED ERROR

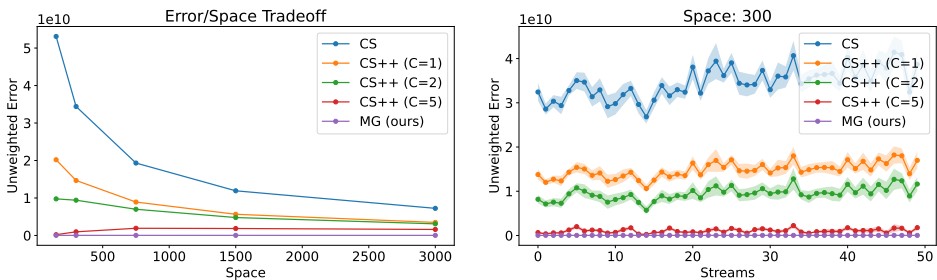

Figure 17: Frequency estimation on the CAIDA dataset with unweighted error and learned predictions.

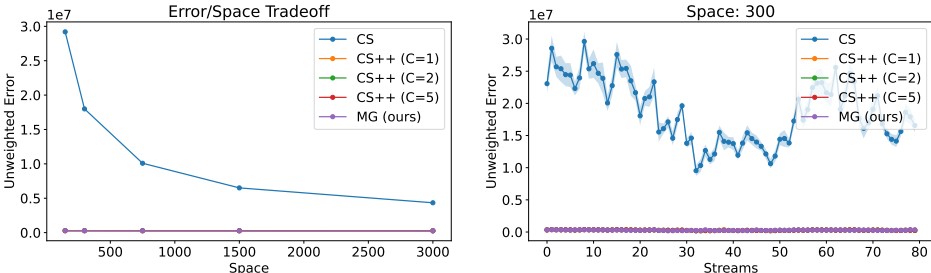

Figure 18: Frequency estimation on the AOL dataset with unweighted error and learned predictions.

