# OpenReview forum: "Learning-Augmented Frequent Directions"
_ICLR.cc/2025/Conference — ICLR 2025 Spotlight_

### Official Review · Reviewer_uku3 · 2024-10-27

**Soundness:** 4
**Presentation:** 4
**Contribution:** 3
**Rating:** 8
**Confidence:** 4

**Summary:**

This work revisits the problem of learning-augmented frequency estimation and provides simplified, improved, and generalized results. In the classic frequency estimation problem, we wish to estimate the number of occurrences of an item in a stream up to some small error. In the “learning-augmented” setting, we assume that some oracle tells us the most frequent items, which allows for better algorithms. Furthermore, the stream is also assumed to follow a Zipfian distribution, that is, there is a clear choice of “most frequent items” so this assumption is actually useful. Prior work on this subject, such as the work of Hsu et al. (2019), uses randomized algorithms based on CountMin, CountSketch, and related constructions. The current work shows that the Misra-Gries algorithm, a classic deterministic algorithm for frequency estimation, actually obtains sharp guarantees for this problem. Furthermore, this work also generalizes this to the vector setting, where the stream items are vectors and the desired output is estimates to the singular vectors of the stream viewed as a matrix. In this setting, Misra-Gries corresponds to the frequent directions algorithm, and the authors obtain similar guarantees.

**Strengths:**

The paper is remarkably well written. Every time I wanted to ask a question to the authors, I felt like it was almost immediately answered in the following text. The contribution is also solid, and gives a clean improvement to prior work on this subject. The experiments were also insightful, and showed how an initial estimate of the top singular values can be used as the “learned frequent directions” for use in the learned frequent directions algorithm. In hindsight, this seems pretty similar to related ideas in numerical linear algebra about using a coarse estimate of the top singular components when designing algorithms for matrix estimation tasks, such as https://arxiv.org/abs/2010.09649 and https://arxiv.org/abs/1511.07263. Connections to such works might be interesting to include.

**Weaknesses:**

The only complaint is that the paper is pretty much a combination of existing and known ideas. However, it is a high quality implementation of such a study and adds valuable information to this literature.

**Questions:**

None at this time.

---

> ### Author Response · Authors · 2024-11-19
>
> > In hindsight, this seems pretty similar to related ideas in numerical linear algebra about using a coarse estimate of the top singular components when designing algorithms for matrix estimation tasks, such as https://arxiv.org/abs/2010.09649 and https://arxiv.org/abs/1511.07263. Connections to such works might be interesting to include.
>
> Thank you for pointing out the relevant works. We have updated our references.

---

> > ### Comment · Reviewer_uku3 · 2024-11-26
> >
> > Thank you for the rebuttal, I have nothing further to discuss.

---

### Official Review · Reviewer_CHdX · 2024-10-29

**Soundness:** 4
**Presentation:** 4
**Contribution:** 4
**Rating:** 8
**Confidence:** 3

**Summary:**

This paper studied the learning-augmented frequent directions, which is a natural generalization of the frequency estimation problem in a stream of elements. The frequency estimation problem asks the following question: we are given a stream of $n$ items that below to at most $d$ distinct elements, what is the additive error we could get for the estimation with a limited memory $m\ll \min(n,d)$. Learning-augmented frequency estimation, which additionally assumes a (potentially erroneous) oracle that returns a predicted frequency of each element *and* the Zipfian distribution of the elements, has been extensively studied in recent years. The state-of-the-art error bound was given by ACNSV [NeurIPS’23] for frequency estimation.

This paper asked if we want to generalize the frequency estimation in high-dimensional applications. Here, a natural setup is to have multiple $d$-dimensional vectors $A_1$, $\cdots$, $A_n$. Furthermore, the following aspects are defined for the generalization.
- To generalize the notion of frequency estimation error, the paper used a notion of the error of vector projection to the singular vectors of the matrix formed by $A=[A^T_1, A^T_2, \cdots, A^T_n]^T$.
- To generalize the distribution assumption of Zipfian distributions, the paper used the notion of the square of the $i$-th largest singular values of the matrix $A$ being inversely proportional to $i$.
- To generalize the frequency prediction, the paper introduced an oracle that directly predicts orthogonal frequency directions, which are the ``right rows’’ an optimal algorithm should have used.

The paper showed that (learning-augmented) frequency estimation can be reduced to the (learning-augmented) frequent direction problem defined above. This justifies the claim that this problem is a *natural* extension of the frequency estimation counterpart.

The paper obtained a frequent direction algorithm with $O(\log{m}\cdot \frac{\log{d/m}}{\log^{2}{d}}\cdot \frac{\||A\||^2_{F}}{m})$ additive error for $m$-memory algorithms even without predictions (by taking advantage of the ‘Zipfian’ distribution of the matrix).  For algorithms with predictions, the error bound becomes $O( \frac{1}{\log^{2}{d}}\cdot \frac{\||A\||^2_{F}}{m})$, assuming the prediction is at least accurate on a constant fraction of the predicted frequency directions. The paper further generalized their results to the frequency estimation problem, showing that it is possible to recover the best bound for the frequency estimation problems using the new algorithm.

**Strengths:**

My overall opinion of this paper is quite positive: it is well-motivated and well-written, and it contains quite some results that could be interesting to a wide range of audiences both in machine learning and algorithms in general. Barring some lower-order exposition issues, the paper is generally easy to follow, and the technical idea, while simple, is quite neat. As such, the paper is a clear acceptance from my perspective.

**Weaknesses:**

I do not see any major weakness in the paper. Some comments for expositions:
- The column indexing of the SVD in algorithm 1 is not as clear as it could be. I suggest having a notation section that lists all the notations you use in the paper.
- Calling the lemma statement of Liberty [Arxiv’22] a ‘fact’ is perhaps overly trivializing the significance of the lemma. I would say to call it a ‘proposition’ (I know it’s a one-line proof, but still :)).
- The discussion about the robustness in the main paper is perhaps too sketchy. I understand that you have a more detailed discussion in the appendix; however, I think it’d be helpful if you could expand the discussion.

**Questions:**

In the experiment, you used SVD as a benchmark. Can you give the error bound for storing all vectors in $A$ and run SVD? In particular, does the error also scale with $\||A\||^2_{F}$?

---

> ### Author Response · Authors · 2024-11-19
>
> > The column indexing of the SVD in algorithm 1 is not as clear as it could be. I suggest having a notation section that lists all the notations you use in the paper.
>
> Thanks for the suggestion. We have added a table of notation to the beginning of the appendix.

---

> ### Author Response · Authors · 2024-11-19
>
> > Calling the lemma statement of Liberty [Arxiv’22] a ‘fact’ is perhaps overly trivializing the significance of the lemma. I would say to call it a ‘proposition’ (I know it’s a one-line proof, but still :)).
>
> Thanks for the suggestion, we agree and have changed it.

---

> ### Author Response · Authors · 2024-11-19
>
> > The discussion about the robustness in the main paper is perhaps too sketchy. I understand that you have a more detailed discussion in the appendix; however, I think it’d be helpful if you could expand the discussion.
>
> We agree. Please see the discussion with reviewer rWSV who made a similar point. We will expand upon this in the main text of the final version.

---

> ### Author Response · Authors · 2024-11-19
>
> > In the experiment, you used SVD as a benchmark. Can you give the error bound for storing all vectors in A and run SVD? In particular, does the error also scale with ||A||_F^2?
>
> The SVD benchmark of storing all the vectors in A and then outputting the optimal rank m approximation can be shown to have error $\Theta\left( \frac{ ||A||_F^2}{m (\log d)^2} \right)$ for our weighted error metric, which is asymptotically what our learned algorithm (with a perfect oracle) obtains. The SVD benchmark does not use predictions but requires $\Omega(nd)$ space, so it is not a sensible streaming algorithm.

---

> ### Comment · Reviewer_CHdX · 2024-11-21
>
> Thanks for the feedback, and in particular confirming the error bound for running SVD with all vectors. I have no further questions.
>
> I also went across my colleagues' reviews. I am not as concerned about having multiple types of predictions, but it might be an interesting point to discuss.
>
> Finally, a small note on style: please have a unified format of references, i.e, either name et al. (conf year) or (name et al. conf year). Right now, both formats are used in the abstract. Please make sure the format is consistent (and without using \cite{} in the abstract, of course).

---

### Official Review · Reviewer_TCrD · 2024-11-03

**Soundness:** 2
**Presentation:** 2
**Contribution:** 3
**Rating:** 6
**Confidence:** 3

**Summary:**

This paper presents a theoretical and empirical study of learning-augmented streaming algorithms, focusing on frequency estimation and its extension to matrix streaming. The authors introduce a learning-augmented variant of the Frequent Directions algorithm and demonstrate both theoretical improvements and practical benefits when combining learned predictions with traditional streaming algorithms.

**Strengths:**

1. The paper provides a clear theoretical analysis of both learning-augmented Misra-Gries and Frequent Directions algorithms. Achieves state-of-the-art bounds for learning-augmented frequency estimation with a simpler, deterministic algorithm

2.  Successfully extends learning-augmented techniques from frequency estimation to matrix streaming

3. Comprehensive experimental evaluation on real-world datasets. Shows 1-2 orders of magnitude improvement over baseline algorithms and very close to SVD algorithm which is memory intensive.

**Weaknesses:**

1. Limited diversity in the prediction models used. More comparison with other learning-augmented streaming algorithms would strengthen the evaluation


2. Could benefit from ablation studies on prediction quality vs. performance

**Questions:**

1. Is there a fundamental trade-off between space usage and prediction accuracy?
2. How does the performance vary with different stream lengths?
3. How does the approach scale to very high-dimensional data?

---

> ### Author Response · Authors · 2024-11-19
>
> > Limited diversity in the prediction models used.
>
> We are using the most popular predictions used in prior learning augmented frequency estimation works.

---

> ### Author Response · Authors · 2024-11-19
>
> > More comparison with other learning-augmented streaming algorithms would strengthen the evaluation
>
> We compare against the best performing algorithms from prior learning-augmented frequency estimation papers: learning-augmented CS from Hsu et al. and learning-augmented CS++ from Aamand et al. For frequent directions, we propose the first learning augmented algorithm and there are therefore no natural alternatives to compare to, although such alternatives would be very interesting.

---

> ### Author Response · Authors · 2024-11-19
>
> > Could benefit from ablation studies on prediction quality vs. performance
>
> Thanks for this comment, this is an important consideration which we did not emphasize enough in the draft. Our existing experiments indirectly show the performance of our algorithm as the prediction quality varies, for both frequency estimation and frequent directions. Our datasets consist of streams which vary with time. We used the earliest data stream to construct our predictor and evaluated the performance of our algorithms across all the later data streams. For example in Figures 2 (a) and 2(b) (the right plots), we use predictions based on the first day’s stream for all the subsequent days. In these plots, as the x-coordinate increases, the prediction becomes increasingly stale since it is obtained from a stream further and further in the past. Nevertheless, the performance of our algorithms remains the best among all the baselines, even as the quality of the predictions degrades.
>
>
> We took up your helpful suggestion of doing a more direct analysis of how noise affects our algorithm. Please see Figure 6 in the appendix of the revised draft. We artificially added iid Gaussian noise of varying scales to the projection matrix which constitutes our prediction. The figure demonstrates that the performance of our algorithm degrades as some power of the scale of the error.

---

> ### Author Response · Authors · 2024-11-19
>
> > Is there a fundamental trade-off between space usage and prediction accuracy?
>
> We are not sure exactly what you mean by prediction accuracy. Here are a few comments on tradeoffs: please let us know if you are asking about something else.
>
> * Regarding tradeoffs between space usage and the error of our algorithms, our main theorems and experiments directly address this tradeoff.
>
> * Our theoretical and experimental results demonstrate that learned predictions yield algorithms with improved tradeoffs between space usage and error. In this sense, for a fixed utility, accurate predictions allow for less space.
>
> * Another tradeoff is the space required to store the prediction model. Though an interesting question, this is not a focus of our work. Rather, our goal is to optimize the space/error tradeoff of streaming algorithms given a black box predictor.

---

> ### Author Response · Authors · 2024-11-19
>
> > How does the approach scale to very high-dimensional data?
>
> The space of our algorithm has *no* dependence on $n$, the number of vectors that are arriving in the stream. It only depends on $m$, a user specified number of rows that we want to keep track of.  Since the stream consists of $d$ dimensional vectors, we also maintain $m$ vectors in $d$ dimensions. Overall, our algorithm only has a linear dependence on the dimension $d$, in both time and space. This is the best one can hope for, because even reading one vector in the stream takes $\Omega(d)$ time and storing even one singular direction takes $\Omega(d)$ space. Our experiments also demonstrate that our algorithm can scale to large dimensions since we test on datasets with dimensionality up to 1920.

---

> ### Author Response · Authors · 2024-11-24
> **Follow up to Reviewer TCrD**
>
> Dear Reviewer TCrD,
>
> Thank you again for your review.
>
> We believe we have thoroughly addressed your main concerns about diversity of prediction models and clearly outlined how our approach scales with different stream lengths and to high dimensional data. If any of your concerns have not been addressed, could you please let us know before the end of the discussion phase?
>
> Many thanks,
> The authors

---

> > ### Comment · Reviewer_TCrD · 2024-11-26
> >
> > Thank you for addressing my questions and I have read all other reviews too. I decided to increase my score to 6.

---

### Official Review · Reviewer_rWSV · 2024-11-03

**Soundness:** 4
**Presentation:** 3
**Contribution:** 3
**Rating:** 8
**Confidence:** 4

**Summary:**

The paper discusses the problem of online learning of frequent directions given a stream of vectors. In particular, they also consider the learning-augmented problem, where one has access to a predictor that provides estimate of which directions are frequent. They also address the problem of frequent items, since this a special case of the frequent direction problem, where item i is associated to the standard vector e_i.

The contribution of the author is two-fold. They first study the problem of frequent directions using power-law distributed data (i.e., in this case the singular values of the input matrix A, that is given online row by row, follow a power law or Zipfian distribution). To this goal, they analyze the Frequent Direction Algorithms (Ghashami et al., 2016) in this setting (Theorem 3.3). Additionally, they also shows that if one has access to a perfect learning oracle, the expected error of the algorithm can decrease. As a special case, they also shows that a variation of this algorithm (based on the Misra-Gries algorithm, that is the 1-dimensional variant of the frequent direction algorithm), allows to solve the (learning-augmented) frequent item problem with Zipfian distribution law of the frequency. In particular, they retrieve the results of the previous work with a simple deterministic algorithm, compared to previous work that is based on CountMin sketch and is randomized.

**Strengths:**

The paper is within the algorithms with prediction framework, that was subject to a considerable amount of research in the last years. The paper studies a new problem in the algorithms with prediction framework: learning-augmented frequent directions.

One of the main strength of the algorithm is that they are also able to recover, with a simple algorithm, results of the previous work for the special case of frequent items.

Another strength of the paper is that it is extremely well-written. The technical section is easy to follow. The results look correct.

Due to the simplicity of the algorithm, clarity of exposition, and the relevance of the results, I believe this paper provides a good contribution.

**Weaknesses:**

The paragraph on robustness 361-366 is not clear. It seems that they constraint the oracle to provide still a lot of valuable information. I believe it would be better to provide instead sufficient conditions required on the oracle to still obtain the approximation guarantees of Thm 3.4 (or resp., Thm 3.2). It seems to me that appendix E shows that one can accomodate an arbitrary bad oracle (Theorem E.1) using a modification of the algorithm. Unless I am missing something, I believe this is more important than the robustness paragraph 361-366, and should be maybe included in the main paper.

Pages 15-28 contains a sequence of plots with virtually almost no-text (Figure 53 seems a very high number of figures). I think it would be better to select some plots that are then discussed with text to express a message, or summarize the information in a different and more concise manner.

**Questions:**

See weakness.

The lower bound of Thm.1  and Thm.3 seems specific to the algorithm that is used. Is there any lower bound that applies for any algorithm, given an input sequence that satisfies the Zipfian law assumption?

---

> ### Author Response · Authors · 2024-11-19
>
> > The paragraph on robustness 361-366 is not clear ...... I believe this is more important than the robustness paragraph 361-366, and should be maybe included in the main paper.
>
> Thanks for pointing this out. ‘Robustness’ was not a good choice of word for this paragraph since it usually relates to retaining the worst case guarantees from the unlearned setting. Instead, the paragraph illustrates one possible relaxation of the assumption that the oracle is perfect where we still retain the approximation guarantees in Theorem 3.4. The reviewer is correct that Appendix E uses a modification of the Learned Frequent Directions algorithm to accommodate arbitrarily bad oracles (robustness) and we agree that this is important to discuss in the main body. We will update the paper accordingly.

---

> ### Author Response · Authors · 2024-11-19
>
> > Pages 15-28 contains a sequence of plots with virtually almost no-text (Figure 53 seems a very high number of figures). I think it would be better to select some plots that are then discussed with text to express a message, or summarize the information in a different and more concise manner.
>
> Thanks for the comment, we have made the appendix experiments more concise (and moved them to the end of the appendix) in the updated draft.

---

> ### Author Response · Authors · 2024-11-19
>
> > The lower bound of Thm.1 and Thm.3 seems specific to the algorithm that is used. Is there any lower bound that applies for any algorithm, given an input sequence that satisfies the Zipfian law assumption?
>
> This is an interesting open problem. In fact, general lower bounds are currently not known even for the simpler case of learned frequency estimation with the Zipfian law assumption under weighted estimation error.

---

> > ### Comment · Reviewer_rWSV · 2024-11-26
> >
> > I thank the reviewer for their response and for updating their manuscript. After also reading the other reviews, I would like to keep my positive score.

---

### Author Response · Authors · 2024-11-19

We thank the reviewers for their valuable feedback. Answers are given in a response to each review. We have uploaded a revised version of the paper.

---

### Meta-Review · Area_Chair_GBt5 · 2024-12-09

**Metareview:**

This paper presents a learning-augmented version of the Frequent Directions algorithm for matrix streaming. It extends the learning-augmented frequency estimation problem to high-dimensional data, showing significant improvements in both theoretical guarantees and practical performance by combining learned predictions with traditional algorithms.

Reviewers praised the paper for its clarity, strong theoretical contributions, and successful experimental validation. Overall, the paper is seen as a solid and valuable contribution to the field.

**Additional Comments On Reviewer Discussion:**

The reviewers suggested expanding the comparison with other algorithms, adding ablation studies, and exploring different prediction models to further enhance the analysis, and the authors provided detailed responses during the discussion phase.

---

### Decision · Program_Chairs · 2025-01-22

Accept (Spotlight)